



# Air quality model assessment in city plumes of Europe and East Asia

Adrien Deroubaix[1,2], Marco Vountas[1], Benjamin Gaubert[3], Maria Dolores Andrés Hernández[1],
Stephan Borrmann[4], Guy Brasseur[2,3], Bruna Holanda[4], Yugo Kanaya[5], Katharina Kaiser[4], Flora Kluge[6],
Ovid Oktavian Krüger[4], Inga Labuhn[7], Michael Lichtenstern[8], Klaus Pfeilsticker[6], Mira Pöhlker[4],
Hans Schlager[8], Johannes Schneider[4], Guillaume Siour[9], Basudev Swain[1], Paolo Tuccella[10], Kameswara
S. Vinjamuri[1], Mihalis Vrekoussis[1], Benjamin Weyland[6], and John P. Burrows[1]

[1]Institute of Environmental Physics, University of Bremen, Bremen, Germany
[2]Max Planck Institute for Meteorology, Hamburg, Germany
[3]Atmospheric Chemistry Observations and Modeling, National Center for Atmospheric Research, Boulder, CO, USA
[4]Max Planck Institute for Chemistry, Mainz, Germany
[5]Research Institute for Global Change, Japan Agency for Marine-Earth Science and Technology, Yokohama, Japan
[6]Institute of Environmental Physics, University of Heidelberg, Heidelberg, Germany
[7]Climate Lab, Institute of Geography, University of Bremen, Bremen, Germany
[8]Deutsches Zentrum für Luft-und Raumfahrt, Oberpfaffenhofen, Germany
[9]Univ Paris Est Creteil, Université Paris Cité, CNRS, LISA, F-94010 Créteil, France
[10]University of L'Aquila, Department of Physical and Chemical Sciences, L'Aquila, Italy

**Correspondence:** Adrien.Deroubaix@iup.physik.uni-bremen.de

**Abstract.**

An air quality model ensemble is used to represent the current state-of-the-art in atmospheric modeling, composed of two global forecasts and two regional simulations. The model ensemble assessment focuses on both carbonaceous aerosols, *i.e.* black carbon (BC) and organic aerosol (OA), and five trace gases during two aircraft campaigns of the EMeRGe (Effect of

5 Megacities on the Transport and Transformation of Pollutants on the Regional to Global Scales) project. These campaigns, designed with similar flight plans for Europe and Asia, along with identical instrumentation, provide a unique opportunity to evaluate air quality models with a specific focus on city plumes.

The observed concentration ranges for all pollutants are reproduced by the ensemble in the various environments sampled during the EMeRGe campaigns. The evaluation of the air quality model ensemble reveals differences between the two cam-

10 paigns, with carbon monoxide (CO) better reproduced in East Asia, while other studied pollutants exhibit a better agreement in Europe. These differences may be associated to the modeling of biomass burning pollution during the EMeRGe Asian campaign. However, the modeled CO generally demonstrates good agreement with observations with a correlation coefficient (R) of $\approx 0.8$. For formaldehyde (HCHO), nitrogen dioxide ($NO_2$), ozone ($O_3$) and BC the agreement is moderate (with R ranging from 0.5 to 0.7), while for OA and $SO_2$ the agreement is weak (with R ranging from 0.2 to 0.3).

15 The modeled wind speed shows very good agreement (R $\approx 0.9$). This supports the use of modeled pollutant transport to identify flight legs associated with pollution originating from major population centers targeted among different flight plans. City plumes are identified using a methodology based on numerical tracer experiments, where tracers are emitted from city centers. This approach robustly localizes the different city plumes in both time and space, even after traveling several hundred



kilometers. Focusing on city plumes, the fractions of high concentration are overestimated for BC, OA, HCHO, and $SO_2$,
which degrades the performance of the ensemble.

This assessment of air quality models with collocated airborne measurements provides a clear insight into the existing limitations in modeling the composition of carbonaceous aerosols and trace gases, especially in city plumes.

# 1 Introduction

Modeling air quality in megacities or major population centers poses several challenges due to highly variable pollutant emissions resulting in complex atmospheric chemistry in these environments (e.g., Monks et al., 2015; Baklanov et al., 2016). Despite these challenges, accurate air quality modeling is required to assess and provide early warning of the health impacts (e.g., Gurjar et al., 2010). The transport of pollution plumes from these centers also has far-reaching consequences on regional air quality (e.g., Monks et al., 2015), global climate (e.g., Folberth et al., 2015), as well as human and ecosystem health (e.g., Manisalidis et al., 2020).

Air quality models have largely been assessed by comparison with the observations from continent-wide measurement networks (e.g., Tuccella et al., 2012). The European operational air quality forecasting center now uses daily assessments, incorporating observations from measurement networks and satellite columns of CO, $NO_2$, $O_3$ and AOD (Huijnen et al., 2019; Garrigues et al., 2022; Wagner et al., 2021). Aircraft measurements provide the only in-situ data source to evaluate the variability of vertical profiles of atmospheric composition across diverse environments. However, they offer only instantaneous snapshots of atmospheric composition. The variability in modeled concentrations has been often evaluated using an aircraft measurement campaign in a single region, focusing on a unique air quality model, and targeting specific atmospheric components (e.g., Fast et al., 2009; Chen et al., 2020; Menut et al., 2015; Hodzic et al., 2020). Only a few studies have gathered airbone observations in order to evaluate air quality models (e.g., Pai et al., 2020; Wang et al., 2020).

Several model intercomparisons have identified recurrent modeling issues related to emissions, $O_3$ chemistry, and secondary aerosol formation, investigated in regions including Europe, North America, and Asia (e.g., Solazzo et al., 2012; Im et al., 2015a, b; Bessagnet et al., 2016; Mircea et al., 2019; Chen et al., 2019; Li et al., 2019). While some studies have employed airborne observations to evaluate an ensemble of air quality models (e.g., Park et al., 2021), there remains a need to assess an ensemble of modeled concentrations for different regions using consistent aircraft instrumentation. The EMeRGe aircraft campaigns (Effect of Megacities on the Transport and Transformation of Pollutants on the Regional to Global Scales) conducted in Europe (2017) and East Asia (2018) are particularly valuable for studying aerosol and trace gas compositions. For the two campaigns, the German research aircraft, called HALO (High Altitude and LOng Range Research Aircraft), has the same instrumental payload with a focus on city plumes (Andrés Hernández et al., 2022; Förster et al., 2023; Lin et al., 2023).

An identification of city plumes has been proposed for the EMeRGe campaigns by using backward trajectories focusing on the flight legs in the planetary boundary layer (PBL) (Förster et al., 2023). Another modeling approach is possible by releasing tracers (*i.e.* additional numerical gaseous non-reactive species) that are transported by the dispersion model (emitted at the location of the major population center of a given region). This approach can be implemented in online coupled meteorological





and air quality models, alongside all other chemically reactive species (with little additional computational cost, since it does not require calculation by the chemistry scheme). Using the WRF-CHIMERE model (Menut et al., 2021) during the DACCIWA campaign (Knippertz et al., 2017), the transport of pollution from major population centers of the Guinean coast has been investigated with this approach in order to distinguish the anthropogenic pollution from the long-range transport of pollution from biomass burning (Flamant et al., 2018; Deroubaix et al., 2019; de Coëtlogon et al., 2023).

This study uses an ensemble of two regional simulations and two global forecasts to assess modeled concentrations of carbonaceous aerosols, *i.e.* black carbon (BC) and organic aerosol (OA), and trace gases (CO, HCHO, $NO_2$, $O_3$, and $SO_2$) during the EMeRGe campaigns (Section 2). The novelty lies in the simultaneous assessment of these models for two distinct regions using identical aircraft instrumentation, with a specific focus on city plumes. The model ensemble assessment encompasses concentration range analyses and evaluations of statistical performance, considering different sampling times of airborne observations and comparing the results from Europe and East Asia (Section 3). Additionally, the evaluation is focused on city plumes (Section 4). Finally, the performance of the model ensemble is assessed with respect to its ability to identify the similarities in the two regions and to reproduce the composition of city plumes (Section 5).

## 2 Air quality observation and modeling

In this section, we briefly present the datasets used from the two EMeRGe campaigns obtained from the instrumentation on board the HALO aircraft (Section 2.1), and the four simulations used to compose the air quality model ensemble (Section 2.2).

### 2.1 EMeRGe flights and instrumental payload

The two EMeRGe aircraft campaigns took place (1) in Europe during the period 11 – 28 July 2017, and (2) in East Asia during the period 8 March – 9 April 2018. The two campaigns were dedicated to the investigation of the transport and chemical processes that occurs in city plumes. We analyze (1) seven flights in Europe (based at the DLR hangar at Oberpfaffenhofen, Germany), and (2) ten flights in East Asia (based at the Air Asia hangar in Tainan, Taiwan) performed by the HALO research aircraft. Most of the flights were conducted in the lower troposphere (Figure 1). Although the objectives of the two campaigns were the same, the HALO aircraft flew mainly over the ocean in East Asia (due to restrictions in the flight permission over China), whereas in Europe it flew more often over land. This difference may be of importance in the interpretation of the air masses sampled.

From the large set of instruments for which more details are given in Andrés Hernández et al. (2022), we use the measurements of specific aerosol components and trace gases that are relevant to study city plumes. We focus on carbonaceous aerosol because we expect significant proportion of primary BC and OA in city plumes. For trace gases, we analyze the most readily observable by satellites (CO, HCHO, $NO_2$, $O_3$, and $SO_2$). In addition, wind speed is analyzed for its importance on determining pollutant transport. This selection of trace gases is motivated by the intention to assess the proportional relationships between carbonaceous aerosol and these five trace gases, whose columns are also retrieved by remote sensing measurements from satel-



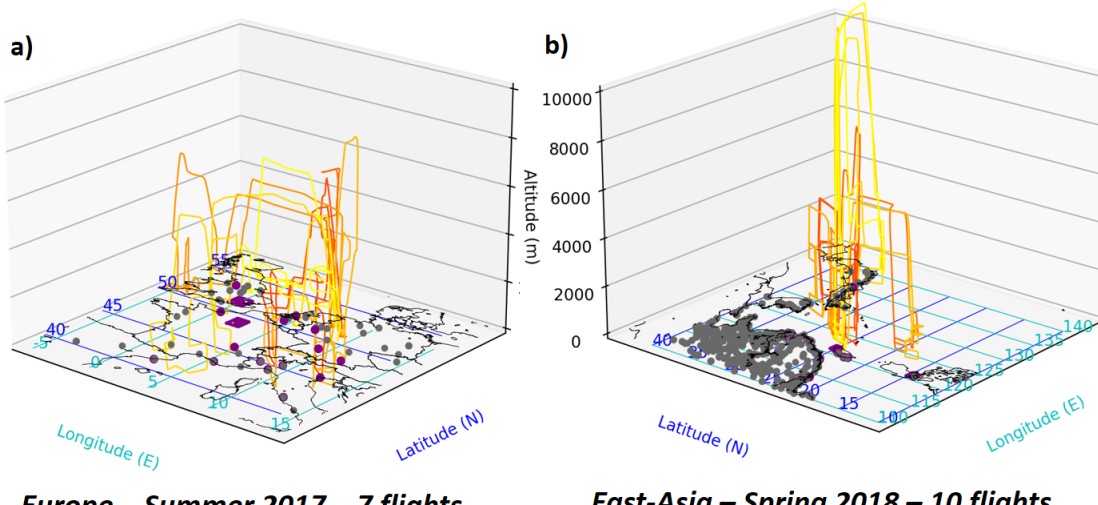

**Figure 1.** *Map of the flights studied of the two EMeRGe campaigns: (a) 7 flights were in Europe during the summer of 2017, and (b) 10 flights were in East Asia during the spring of 2018. The different flights are shown in different colors (from yellow to dark orange), with the altitude represented by a third dimension on the maps. Cities with more than 500,000 inhabitants are displayed (gray dots). The cities of interest in this study are displayed with purple dots.*

lite instrumentation. Consequently, the following measurements of meteorology, trace gases and aerosol concentrations are selected:

– Wind speed (using standard HALO core instrumentation),

– CO and $O_3$ (using the AMTEX instrument based on UV-photo-fluorimetry),

– $NO_2$ and HCHO (using a mini-DOAS instrument based on Differential Optical Absorption Spectrometry),

– $SO_2$ (using a CI-ITMS instrument: Chemical Ionization Ion Trap Mass Spectrometry),

– BC (using the SP2 instrument: Single Particle Soot Photometry),

– OA (using a CTOF-AMS instrument: Time of Flight- Aerosol Mass Spectrometry)

For the European campaign, we analyze about 43 hours of sampling, and about 73 hours for the Asian campaign. The number of observations (at a 1-min averaging time step) for the two campaigns is 6986 for wind speed (no missing values), 6870 for CO, 4751 for HCHO, 5587 for $NO_2$, 6949 for $O_3$, 5360 for $SO_2$, 6412 for BC and 5875 for OA. The instrumental payload of the aircraft enables the same analytical techniques to be applied to the observations made during the two campaigns, and to

assess the similarities and differences of the atmospheric composition observed in the two regions.



## 2.2 Air quality model ensemble

**Table 1.** *Air quality model configurations: the model ensemble is composed of two global forecasts (CAMchem–CESM2 and CAMS–forecast) and two regional simulations using the WRFchem model.*

| Institution | NCAR | ECMWF | IUP |
|---|---|---|---|
| **Model** | CAMchem–CESM2.1 | IFS–CAMS | WRFchem (version 4.3.3) |
| *Domain* | | | |
| **Horizontal resolution** | 0.9 x 1.25° | 40 km | 10 km |
| **Vertical levels** | 56 | 137 | 37 |
| **Output frequency** | 6h | 3h | 1h |
| *Emission* | | | |
| **Anthropogenic** | CMIP6 | CAMS-GLOB-ANTv4.2 | CAMS-GLOB-ANTv4.2 |
| | (Feng et al., 2020) | (Granier et al., 2019) | (Granier et al., 2019) |
| **Biogenic** | MEGANv2.1 | MEGANv2.1 | MEGANv2.1 |
| | (Guenther et al., 2006) | (Guenther et al., 2006) | (Guenther et al., 2006) |
| **Fires** | QFED | CAMS-GFASv1.4 | FINNv1.5 |
| | (Darmenov et al., 2015) | (Inness et al., 2022) | (Wiedinmyer et al., 2011) |
| *Gas and aerosol* | | | |
| **Chemical mechanism** | MOZART4-T1 | CB05 | MOZART4-T1 |
| | (Emmons et al., 2020) | (Inness et al., 2019) | (Emmons et al., 2010) |
| **Aerosol scheme** | MAM4-VBS | IFS-AER | GOCART |
| | (Tilmes et al., 2019) | (Rémy et al., 2019) | (Chin et al., 2002) |
| **Boundary conditions** | none | none | CAMchem–CESM2.1 |

There are different approaches to setting up an air quality model ensemble, depending on the analyses planned to answer a specific scientific question. In this study, we set up a model ensemble representing the state-of-the-art in atmospheric composition modeling to investigate the observed and modeled atmospheric composition in city plumes sampled during the EMeRGe 100 campaigns.

The air quality model ensemble is composed of regional simulations performed with the WRFchem model (Grell et al., 2005; Fast et al., 2006; Powers et al., 2017) and two global simulations (Table 1). In order to analyze the influence of the meteorological data driving the regional simulation, two high resolution (*i.e.* 10 km) simulations are performed for each of the two regions using two different sets of meteorological input data.

The simulation performed with the final operational global analysis (FNL) produced by the Global Data Assimilation System of the US National Center for Environmental Prediction (NCEP–FNL) (NCEP, 2022) is referred to as (1) WRFchem–FNL. The simulation performed with the ERA5 reanalysis (Hersbach et al., 2020) produced by the European Centre for Medium-Range



Weather Forecasts (ECMWF–ERA5) is referred to as (2) WRFchem–ERA5. Additionally, two global simulations are selected. The global atmospheric forecast provided by ECMWF through the Copernicus Atmosphere Monitoring Service is referred to

as (3) CAMS–forecast, and the global simulation provided by the US National Center for Atmospheric Research using the Community Atmosphere Model with Chemistry (Buchholz et al., 2019) is referred to as (4) CAMchem–CESM2 .

The main differences between the model configurations chosen for the four simulations are (i) the modeled domain setup, (ii) the emission datasets, and (iii) the chemistry and aerosol schemes (Table 1). The model configurations are related because both regional simulations use CAMchem–CESM2 as boundary conditions, and CAMS-GLOB-ANTv4.2 (Granier et al., 2019)

as anthropogenic emissions provided by ECMWF. In addition, both regional simulations use either NCEP–FNL as input meteorological datasets, which is also used by CAMchem–CESM2, or the ECMWF–ERA5 reanalysis, which is based on the ECMWF meteorological model.

The modeled OA corresponds to the sum of primary and secondary organic aerosol. Aerosol concentrations are measured with a cut-off diameter of 1 $\mu$m and the modeled concentrations are compared accordingly.

Based on the horizontal, vertical and temporal resolution of the outputs, the simulations are interpolated in time and in space according to the flight path of the HALO aircraft. In other words, the modeled concentrations are interpolated along the flight positions with a triple interpolation (bilinear horizontally, linear vertically and linear between two time steps), which enables modeled concentrations to be generated at the locations and time steps of the observations made in the HALO aircraft. Since the aircraft is moving at a horizontal speed of approximately 600 km/h (10 km/min), different averaging time steps can affect

the ability of the air quality model ensemble to reproduce the observed concentrations, especially because of the difference in horizontal resolution ranging from 10 km for the regional simulations of WRFchem to about 100 km for the global simulation of CAMchem–CESM2. In order to compare the observations of the HALO aircraft with the four simulations of the air quality model ensemble, different averaging time steps of 1, 3 and 10-min are used.

## 3 Model assessment in the various environments sampled

This section is dedicated to the assessment of the air quality model ensemble by comparison with the measurements performed during the two EMeRGe campaigns in all the various environments sampled. We start the analysis by evaluating the air quality model ensemble in detail for two selected flights because they successfully sampled the plumes of two megacities, one flight from the European campaign sampling the plumes of Paris and London, and one flight from the Asian campaign sampling the plumes of Manila and Taipei (Section 3.1). We then undertake a statistical evaluation of the model ensemble of all the flights

of the two campaigns (Section 3.2) complemented with an analysis of the modeled and observed concentration ranges (Section 3.3).



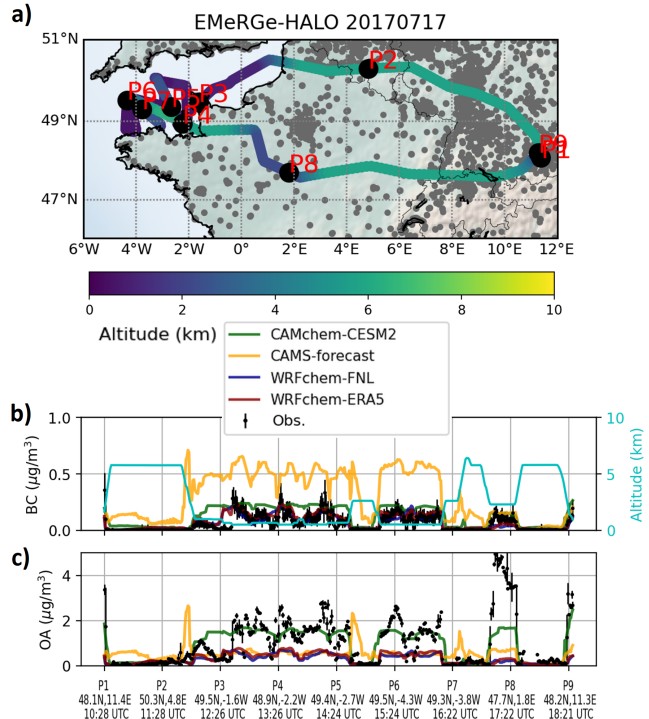

**Figure 2.** *a) Map of the EMeRGe flight on 17 July 2017, and time series of observed and modeled concentrations at a 1-min averaging time step of b) black carbon (BC) and (c) organic aerosol (OA). Observations are presented with the standard deviation of the measurements during the time step (black dots with vertical bars). The air quality model ensemble (colored lines) is composed of two global simulations, CAMchem–CESM2 and CAMS–forecast and two regional simulations, WRFchem–FNL and WRFchem–ERA5. Aerosol concentrations are measured with a cut-off diameter of 1 μm and the modeled concentrations are shown accordingly.*

## 3.1 Two selected flights

### 3.1.1 "English Channel Flight" - 17 July 2017

For the European campaign, we chose to focus on a flight from Munich to the English Channel and back to Munich, planned to

sample the plumes from London and from Paris based on meteorological forecast which predicted mainly southerly transport (Figure 2 and Figure 3). Take-off took place at 10:28 UTC from Munich (waypoint P1) towards the English Channel, flying over Germany and Belgium at around 6 km amsl in the free troposphere.

From the waypoints P1 to P2, the concentrations of short-lived trace gases ($NO_2$, HCHO, $SO_2$) and of carbonaceous aerosols (*i.e.* BC and OA) are low (except during take-off in the airport area). The CO concentration is stable around 80 ppb, while $O_3$

exhibited high variability (50 to 80 ppb). The modeled concentrations agree well between P1 and P2, except for $O_3$, for which the high concentrations (reaching 80 ppb) are not reproduced.



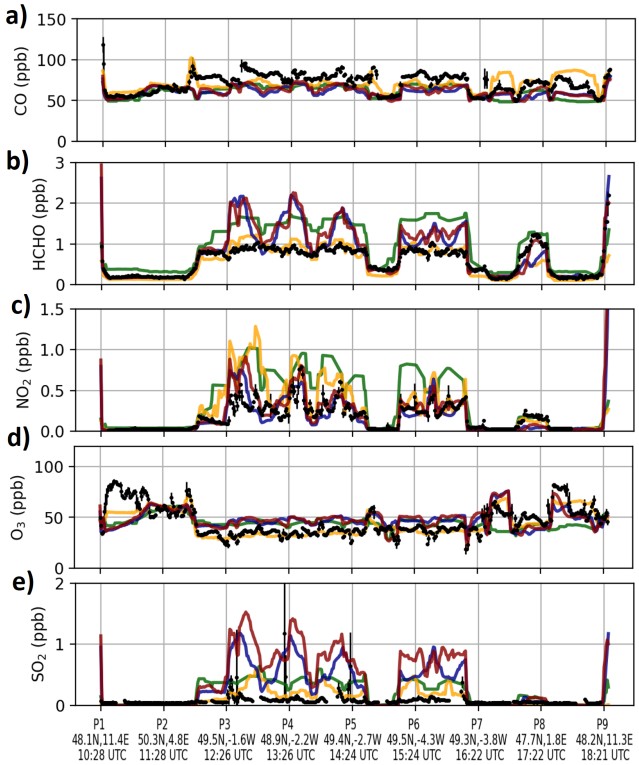

**Figure 3.** *Time series of the EMeRGe flight on 17 July 2017 of observed and modeled concentrations at a 1-min averaging time step of a) carbon monoxide (CO), b) formaldehyde (HCHO), c) nitrogen dioxide (NO$_2$), d) ozone (O$_3$) and e) sulfur dioxide (SO$_2$). Observations are presented with the standard deviation of the measurements during the time step (black dots with vertical bars). The air quality model ensemble (colored lines) is composed of two global simulations, CAMchem–CESM2 and CAMS–forecast and two regional simulations, WRFchem–FNL and WRFchem–ERA5. The color coding is the same as in Figure 2.*

From P2 to P3, the HALO aircraft descends from 6 to 1 km amsl to cross the English Channel to sample the London plume. From P3 to P7, the aircraft performs transects to cross the assumed location of the London plume. The observations of BC and OA show high variability over the English Channel with BC ranging from 0.1 to 0.5 $\mu$g/m$^3$ and OA ranging from 0.5 to 3 $\mu$g/m$^3$. The observed O$_3$ concentration decreases slightly, whereas CO, HCHO, NO$_2$ and SO$_2$ increase. The modeled CO concentration is in good agreement with the observed one, while the O$_3$ and HCHO concentrations are overestimated (by $\approx$ 10 ppb and $\approx$ 0.5 ppb, respectively), except for CAMS–forecast, which is in good agreement for both. The NO$_2$ concentrations modeled by the two regional simulations are in better agreement than the global simulation, which overestimates the NO$_2$. The modeled SO$_2$ concentrations are overestimated by the air quality model ensemble and especially by the two regional simulations. BC is well reproduced, except for CAMS–forecast, which overestimates BC. OA is underestimated, except for CAMchem–CESM2, which reproduces the observed concentration range.




From P7 to P8, the aircraft begins the return leg to Munich, passing south of Paris and flying at low altitude (about 1 km amsl) to sample its pollution plume near P8. By comparing the concentration observed over the English Channel (P3 to P7), we see that the BC concentration is similar, while the OA is about double that for the assumed location of the Paris plume. The concentration ranges are comparable for CO, $O_3$, HCHO and $SO_2$, while $NO_2$ is twice as low for the Paris plume. The modeled concentrations are in good agreement for all the variables between P7 and P8, except for OA, for which the high concentration reaching 4 $\mu g/m^3$ is underestimated by the four simulations, and especially by the regional simulations.

From P8 to P9, the aircraft flies at 6 km amsl, the observed $O_3$ concentration reaches levels comparable to those observed between P1 and P2. The air quality model ensemble is in good agreement with the observations in the free troposphere. High $O_3$ concentration values up to 80 ppb are reproduced by the model ensemble, except for CAMchem–CESM2. Using 3-min and 10-min averaging time steps, the same figures are presented (Figure A1 at 3-min and Figure A2 at 10-min) and the interpretation of the results is unaffected.

### 3.1.2 "Manila Flight" - 28 March 2018

For the Asian campaign, we focus on a flight from Tainan (Taiwan) to the Manila metropolitan area in the Philippines. After take-off, the aircraft flew rapidly to the Philippines, where two flight legs were conducted at low altitude to sample the Manila plume. After completing these legs, the aircraft returned to Taiwan at high altitude. As it approached the Taiwan Island, in order to sample the Taipei plume, it flew at low altitude along the coast from Taiwan to Taipei before turning and flying back to Tainan (Figure 4 and Figure 5).

From P1 to P2, the HALO aircraft flies at high altitude (about 6 km amsl). The observed concentration of BC is less than 0.5 $\mu g/m^3$, while that of OA is greater than 1 $\mu g/m^3$. CO and $O_3$ concentrations are high near the island of Taiwan, reaching 200 ppb for CO and 100 ppb for $O_3$, and the modeled concentrations reproduce the high concentrations near the island. The modeled concentrations are in good agreement with the observations, except that the OA concentration is underestimated.

In P2, the aircraft rapidly decreases altitude to about 2 km, and maintains this altitude until P5 to catch the Manila plume. The observed concentrations of BC and OA increase significantly during three sequences of a few minutes, going from 0.1 to 1.3 $\mu g/m^3$ for BC and from 0.2 to 2.5 $\mu g/m^3$ for OA. The observed CO concentration is also increasing during three sequences of a few minutes, going from 150 ppb to 200 ppb, and that of HCHO from 0.4 to 1 ppb. The observed $O_3$ concentration is stable (within a range between 40 and 60 ppb) and $NO_2$ and $SO_2$ concentrations are low ($\approx$ 0.2 ppb and 0.1 ppb respectively). The last of the three increases (between P3 and P4) is associated with a higher $SO_2$ concentration reaching 0.8 ppb.

The modeled concentrations of BC do not reproduce the three observed increases, whereas the increases in OA reaching 2.5 $\mu g/m^3$ are reproduced by the simulations of the air quality model ensemble. The influence of meteorological input datasets for the regional simulation is noted, with better agreement for OA with ECMWF–ERA5 than with NCEP–FNL. For CO, the concentration of the background level ($\approx$ 150 ppb) is reproduced by CAMS–forecast, whereas the other simulations underestimating CO by $\approx$ 50 ppb. The three increases seen for CO observations are reproduced by the air quality model ensemble, except for CAMchem–CESM2. The air quality model ensemble captures the observed concentration variability in HCHO and $O_3$, while overestimating $SO_2$ and underestimating $NO_2$ for all simulations.



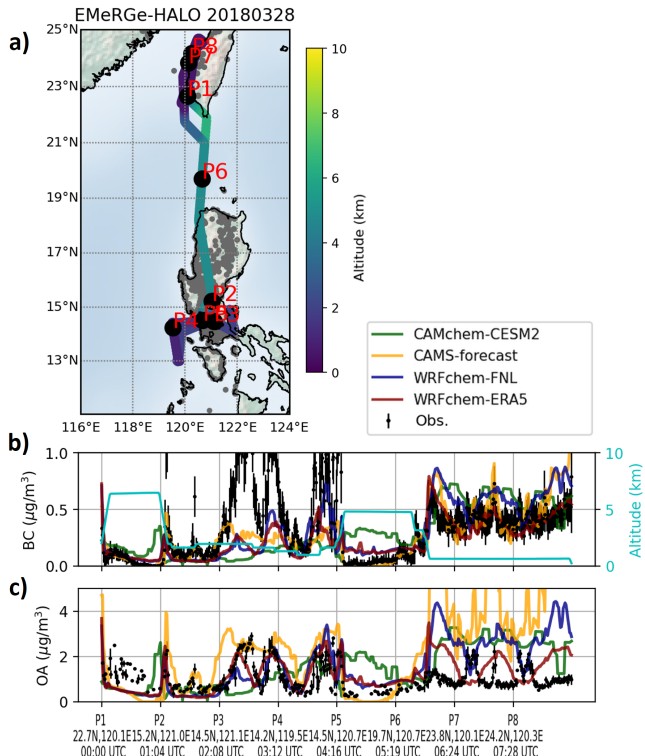

**Figure 4.** *a) Map of the EMeRGe flight on 28 March 2018, and time series of observed and modeled concentrations at a 1-min averaging time step of b) black carbon (BC) and (c) organic aerosol (OA). Observations are presented with the standard deviation of the measurements during the time step (black dots with vertical bars). The air quality model ensemble (colored lines) is composed of two global simulations, CAMchem–CESM2 and CAMS–forecast and two regional simulations, WRFchem–FNL and WRFchem–ERA5. Aerosol concentrations are measured with a cut-off diameter of 1 μm and the modeled concentrations are shown accordingly.*

From P5 to P7, the aircraft returns to Tainan at 6 km altitude until it reaches the coastal area of Taiwan Island. Between P7 and P8, the aircraft flies at 1 km altitude to catch the Taipei plume. When the aircraft is at low altitude, BC and OA concentrations increase to $\approx 0.5$ μg/m$^3$ and $\approx 1$ μg/m$^3$ respectively. For OA, there are clear increases of a few minutes which are not clearly associated with increases in BC. For CO, O$_3$, NO$_2$ and SO$_2$, the concentrations reach higher levels than from P2 to P5 during short increases of a few minutes. HCHO measurements are not available from P6 to the end of the flight.

Along the coastal area of Taiwan Island, the modeled BC concentrations are in good agreement with the observations, while OA is overestimated by the model ensemble. The CO observations are underestimated by the models except for the CAMS forecast. The CAMS forecast overestimates O$_3$, which is in turn well simulated by the rest of the models. It is interesting to note the influence of the meteorological input datasets for the regional simulation between P6 and P8, as the modeled concentration of OA and CO using ECMWF–ERA5 is approximately half that modeled using NCEP–FNL. For NO$_2$ and SO$_2$,





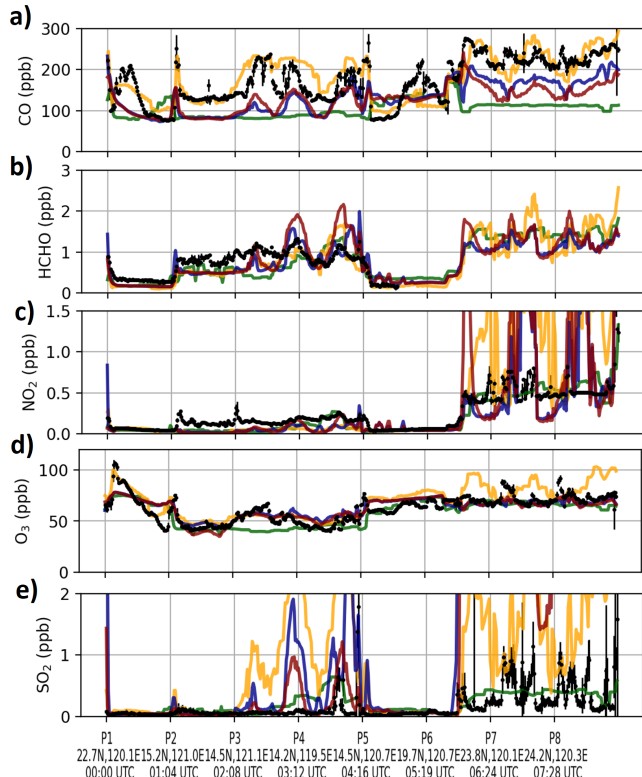

**Figure 5.** *Time series of the EMeRGe flight on 28 March 2018 of observed and modeled concentrations at a 1-min averaging time step of a) carbon monoxide (CO), b) formaldehyde (HCHO), c) nitrogen dioxide (NO$_2$), d) ozone (O$_3$) and e) sulfur dioxide (SO$_2$). Observations are presented with the standard deviation of the measurements during the time step (black dots with vertical bars). The air quality model ensemble (colored lines) is composed of two global simulations, CAMchem–CESM2 and CAMS–forecast and two regional simulations, WRFchem–FNL and WRFchem–ERA5. The color coding is the same as in Figure 2.*

only CAMchem–CESM2 reproduces the observed concentration level, but the short increases are not modeled, while the other three simulations overestimate these increases in trace gases. The increase in averaging time steps does not change significantly the interpretation of the results, , even for CAMchem–CESM2 which has the lowest spatial resolution, as shown in Figure A3 at 3-min and Figure A4 at 10-min averaging time steps

By comparing the concentrations observed in the Manila plume to the Taipei plume, we see that the NO$_2$ and SO$_2$ concentrations are higher in the Taipei plume (0.5 compared to 0.2 ppb and 0.5 compared to 0.1 ppb, respectively), whereas BC and OA concentrations are significantly higher in the Manila plume (0.8 compared to 0.4 $\mu$g/m$^3$ and 2 compared to 1 $\mu$g/m$^3$, respectively). In the study of the "English Channel Flight", we also see from the observations that the relative amounts of BC or OA compared to the five trace gases studied are notably different in the plumes considered to be coming from London and

Paris.



All simulations in the ensemble have identifiable biases for some of the variables. However, each simulation has variables and flight legs for which the agreement with observations is best. The two regional simulations are in good agreement with the observed concentrations of aerosol and trace gases, compared to those simulated by the two global model simulations. The four simulations reproduce the aerosol and trace gases in aged air masses that have been transported several hundred kilometers

from their pollution sources. The agreement between the modeled and observed carbonaceous aerosols and trace gases seems to be poorer in polluted (*i.e.* high-concentration) flight legs, which will be investigated in Section 4.

### 3.2 Model ensemble performance in Europe and East Asia

**Table 2.** *Correlation coefficients (R) between observed and modeled variables by an air quality model ensemble of the four simulations for wind speed, the five studied trace gases (CO, HCHO, NO$_2$, O$_3$, and SO$_2$), as well as black carbon (BC) and organic aerosol (OA). The R-values at a 1-min averaging time step are shown during the EMeRGe campaigns, with those from Europe on the left side of the table (left part) and those from Asia on the right side.*

| | Correlation coefficient (R) | | | | | | | |
|---|---|---|---|---|---|---|---|---|
| **Model** | CAMchem CESM2 | CAMS forecast | WRFchem FNL | | CAMchem CESM2 | CAMS forecast | WRFchem FNL | WRFchem ERA5 |
| | **EMeRGe–Europe - N = 2603** | | | | **EMeRGe–Asia - N = 4383** | | | |
| Wind speed | 0.82 | 0.94 | 0.95 | 0.95 | 0.59 | 0.94 | 0.95 | 0.95 |
| CO | 0.42 | 0.52 | 0.49 | 0.45 | 0.41 | 0.79 | 0.74 | 0.73 |
| HCHO | 0.80 | 0.84 | 0.84 | 0.82 | 0.58 | 0.58 | 0.68 | 0.64 |
| NO$_2$ | 0.46 | 0.49 | 0.69 | 0.65 | 0.45 | 0.56 | 0.59 | 0.70 |
| O$_3$ | 0.69 | 0.77 | 0.69 | 0.68 | 0.38 | 0.54 | 0.48 | 0.48 |
| SO$_2$ | 0.28 | 0.45 | 0.41 | 0.41 | 0.12 | 0.26 | 0.18 | 0.19 |
| BC | 0.58 | 0.47 | 0.63 | 0.59 | 0.36 | 0.46 | 0.60 | 0.61 |
| OA | 0.68 | 0.58 | 0.67 | 0.67 | 0.25 | 0.32 | 0.42 | 0.48 |

The air quality model ensemble is quantitatively assessed by comparison with the observations from all flights of the two EMeRGe campaigns. Statistical metrics are used for this evaluation. These include the Pearson correlation coefficient (R)

between the four simulations and the observations, complemented by the mean bias, root mean square error (RMSE), and linear regression values obtained between the observations and each model. An additional focus is the assessment of the modeling of wind speed due to its important role in pollutant transport.

The evaluation is done by investigating the statistical metrics obtained for each campaign separately (Section 3.2.1) in order to analyze the differences in the performance of the models (for R-values in Table 2 and Table A1, for mean bias in Table A2

and for RMSE in Table A3). The evaluation is done again by investigating the statistical metrics using all flights performed during the two campaigns jointly (Section 3.2.2) to understand the similarities of their performance for each of the variables





**Table 3.** *Correlation coefficients (R) between observed and modeled variables by an air quality model ensemble of the four simulations for wind speed, the five studied trace gases (CO, HCHO, NO$_2$, O$_3$, and SO$_2$), as well as black carbon (BC) and organic aerosol (OA). The R-values at a 1-min averaging time step are shown for all observations of the two EMeRGe campaigns.*

| | Correlation coefficient (R) | | | |
|---|---|---|---|---|
| **Model** | CAMchem CESM2 | CAMS forecast | WRFchem FNL | WRFchem ERA5 |
| | EMeRGe - All observations - N = 6986 | | | |
| Wind speed | 0.69 | 0.94 | 0.95 | 0.95 |
| CO | 0.62 | 0.86 | 0.83 | 0.82 |
| HCHO | 0.59 | 0.66 | 0.71 | 0.69 |
| NO$_2$ | 0.45 | 0.53 | 0.56 | 0.65 |
| O$_3$ | 0.54 | 0.67 | 0.61 | 0.60 |
| SO$_2$ | 0.13 | 0.27 | 0.19 | 0.20 |
| BC | 0.48 | 0.53 | 0.67 | 0.67 |
| OA | 0.32 | 0.24 | 0.31 | 0.36 |

(Table 3). The evolution of the statistical metrics is also studied with longer durations of the averaging time step of 3-min and 10-min.

The differences and similarities of the ensemble performance obtained for each trace gas, BC and OA are identified with R-values, which are indicative of the ability of models to capture the variability of observations, and with slope values indicating the existence of under- or over-estimation of observations. We compare the linear regressions obtained between the observations and the four simulations for the European campaign, for the Asia one, and for all observations of the two campaigns (Figure A6 for BC, Figure A7 for OA, Figure A8 for CO, Figure A9 for HCHO, Figure A10 for NO$_2$, Figure A11 for O$_3$, Figure A12 for SO$_2$ and Figure A13 for wind speed).

### 3.2.1 Performance differences between campaigns

Examining the differences between the two campaigns, we see that the ranges of R-values obtained for the four simulations performed for either Europe or Asia are comparable for all pollutants (Table 2 and Table A1). The R-values for wind speed are greater than 0.9 (except for CAMchem-CESM2, probably due to its low vertical resolution at altitudes higher than 6 km, where the aircraft flew to transit from one city to another). While the highest R-values for trace gases and carbonaceous aerosols are obtained for HCHO in Europe (R $\approx$ 0.8), the other pollutants have lower R-values in both regions. We note that only for CO, the air quality model ensemble performances are lower in Europe (R $\approx$ 0.5) than in Asia (R $\approx$ 0.7, except for CAMchem–CESM2 for which R is $\approx$ 0.4 for the two regions). For OA, the model ensemble performances are better in Europe (R $\approx$ 0.6) than in Asia (R $\approx$ 0.4), while for BC the performance of the model ensemble is similar in the regional simulations (R $\approx$ 0.6) and in



the CAMS–forecast (R ≈ 0.5). For HCHO, $O_3$, $SO_2$, the model ensemble performance is also better in Europe (R ≈ 0.8, 0.7,
and 0.4, respectively) than in Asia (R ≈ 0.6, 0.5, and 0.2, respectively). For $NO_2$, the model ensemble performance is similar
in the two regions, with the R-values for the two regional simulations being higher than for the two global simulations (R ≈
0.7 compared to 0.5).

The bias and RMSE analyses confirm the evaluation of the air quality model ensemble based on the R-values (Table A2
and Table A3). The two regional simulations using two different meteorological input datasets (*i.e.* WRFchem–FNL and
WRFchem–ERA5) have comparable R-values, biases and RMSE, except for $NO_2$ which is best reproduced by WRFchem–
ERA5 (R ≈ 0.7 for WRFchem–ERA5 versus 0.6 for WRFchem–FNL). We note the similarity of the R-values, biases and
RMSE with the longer time steps (data averaged at 3-min and 10-min), which shows that the time step does not affect the
interpretation of the results.

### 3.2.2 Performance similarities between campaigns

Considering the two campaigns jointly (Table 3), the statistical metrics of the air quality model ensemble show good overall
agreement for CO (R ≈ 0.8), and to a lesser extent for HCHO (R ≈ 0.7). We note also that the statistical metrics for CO, BC
and OA are higher than looking at each campaign separately, which is due to larger range of concentration when considering
the two campaigns jointly. For $O_3$, $NO_2$ and BC, the agreement of the models with the observations is moderate (R ≈ 0.6),
while the agreements are weak for OA (R ≈ 0.3) and for $SO_2$ (R ≈ 0.2).

For BC and OA, the slopes of the linear regression of all simulations are less than one in Europe and greater than one in
Asia, resulting in slopes greater than one for the two campaigns jointly, because the highest observed concentrations are in
Asia. For CO, the slopes are less than one in both regions, and CAMS–forecast has the slope closest to one, which is associated
with the best results. For HCHO, the best model performance in Europe is not associated with slopes closer to one, and for the
two campaigns the slopes are close to one, so the high values are well reproduced by the models. For $NO_2$, despite similar R-
values, the slopes are lower than 1 in Europe and higher than 1 in Asia. Except for CAMchem–CESM2, which has the weakest
R-values for the two campaigns associated with slopes closest to 1, showing that the high concentrations are well captured. For
$O_3$, the low R-values obtained in Asia are not associated with slopes different from those obtained in Europe, leading to slopes
close to 1 for the two campaigns and similar concentration ranges for the four simulations.

For $SO_2$, the low R-values obtained in Asia are associated with steep slopes and very high intercepts, indicating overestima-
tion of low and high $SO_2$ concentrations. Furthermore, despite similar R-values for the CAMS–forecast with the other three
simulations, the slope associated with this simulation stands out from the others because it is less than one, so that high $SO_2$
concentrations are better reproduced. The performance of the air quality model ensemble for $SO_2$ is noteworthy because the
mean bias and RMSE are very high and associated with low R-values for the four simulations.

The evaluation of the air quality model ensemble shows that the performance is worse in Asia, where the pollutant concentra-
tions are underestimated. Furthermore, we found that the averaging time step does not affect the performance of the air quality
model ensemble. However, the ranges of the concentrations are important as we study the different environments sampled by
the aircraft. Since this evaluation is made by comparing the observations and the four simulations in space and time, the poor



performance of the air quality model ensemble could hide a good representation of the concentration ranges (*i.e.* leaving the time dimension out of the comparisons).

**3.3    Concentration ranges during the two campaigns**

**Figure 6.** *Observed and modeled concentration ranges for the two EMeRGe campaigns in Europe and in Asia of (a) BC, (b) OA, (c) CO and (d) O$_3$ at 1-min averaging time step.*

We analyze the modeled and observed concentration ranges, focusing on pollutants with lifetimes longer than a day (BC, OA, CO and O$_3$) in order to be able to draw conclusions on the regional differences observed between Europe and Asia (Figure 6). For other species with shorter lifetimes, we cannot draw conclusions at the regional scale because the observations only



reflect the atmospheric composition locally sampled. Therefore, this analysis is limited to the comparison of the modeled and
observed ranges (Figure A5).

Observations of pollutants with longer lifetimes show that a greater fraction of high concentration values are observed in
Asia for BC, OA and CO. The observed ranges of $O_3$ concentrations are similar between Europe and Asia. For CO and $O_3$, the
observed and modeled concentration ranges are in agreement, with the fraction of low concentrations better reproduced by the
CAMS–forecast than by the other simulations, which overestimate them. For HCHO, the range of concentrations in Europe is
well simulated, but the low concentrations in Asia are overestimated (Figure A5). Similarly, for $NO_2$, the low concentrations
are overestimated by the model ensemble in both regions. For $SO_2$, the high concentrations are overestimated by the model
ensemble.

The modeled concentration ranges for BC are well reproduced by the air quality model ensemble in both regions, except
for the CAMS–forecast, for which concentrations are overestimated in Europe (Figure 6). Conversely, the modeled concen-
tration ranges for OA are poorly reproduced because the low concentrations in Europe are overestimated by the two regional
simulations and well reproduced by the global simulations. The opposite is true for Asia, where the low concentrations are
overestimated by the global simulations and well reproduced by the two regional simulations. In addition, the fraction of high
OA concentrations is overestimated in Asia by the four simulations.

In conclusion, the assessment of the air quality model ensemble shows differences between the two EMeRGe campaigns.
The model ensemble performs less satisfactorily in Asia, with pollutant concentrations generally underestimated. CO is better
reproduced in Asia, while other variables are better reproduced in Europe, possibly related to significant fire emissions from
Indochina (Lin et al., 2023). The modification of the averaging time step from 1-min to 3-min or to 10-min does not affect
performance.

Overall, CO shows good agreement with observations, while other variables show moderate (HCHO, $NO_2$, $O_3$, BC) to weak
(OA and $SO_2$) agreement. Although taking into account temporal and spatial variability is a challenge for the ensemble, there
is a reasonable representation of concentration ranges. Wind speed is well represented, suggesting the possible use of modeled
pollutant transport to identify flight legs associated with city plumes.

## 4   Model assessment in city plumes

The methodology presented in this section aims to identify the flight legs associated with pollution plumes originating from
major population centers (hereafter referred to as city plumes), the outflows that are targeted by the different flight plans. The
objective is the identification, rather than attribution of city plumes, enabling the assessment of the air quality model ensemble
within this specific environment during the two EMeRGe campaigns jointly.

The identification methodology follows that used by (Deroubaix et al., 2019; Flamant et al., 2018; de Coëtlogon et al., 2023)
for the analysis of the flights performed during the aircraft campaign of the DACCIWA (Dynamics-Aerosol-Chemistry-Cloud
Interactions in West Africa) project (Knippertz et al., 2017). The methodology is presented in Section 4.1 and applied to the two
selected flights (*cf.* Section 3.1). We then repeat the statistical evaluation of the air quality model ensemble from the previous



section (*cf.* Section 3.2), restricted to the identified flight legs (Section 4.2), complemented with an analysis of the modeled and observed concentration ranges in city plumes (Section 4.3).

## 4.1 City plumes identification applied on the two selected flights

The methodology involves releasing tracers from selected source points corresponding to city centers, which are transported depending on the dispersion simulated by the meteorological model. Tracer emissions are constant over time throughout the simulated period, allowing the assessment of temporal variability of the city tracer concentration solely due to meteorological conditions. Thus, the city tracer concentration represents the intensity of the pollution plumes (*i.e.* in terms of dispersion and age) for a given time and location. The dispersion code from WRF–CHIMERE is utilized because the tracer emission 325 implementation is suitable for this purpose (Menut et al., 2021).

The identification methodology is explained through the four steps below:

1. We start by defining the main cities whose pollution plumes may have been sampled by the HALO aircraft in Europe and East Asia. Our goal is to select only the main cities from the two regions. Although more could be added, we consider that this selection for Europe (Table A4) and East Asia (Table A5) is sufficient. For targeted megacities (London, Paris, 330 Manila, Taipei), we use tracers of five source points instead of one: a first is located in the city center and the others 10 km to the north, east, south and west.

2. Tracers emitted in the different city centers are passive gases (with no chemical reaction and with deposition defined like other gaseous species). Tracer emissions are constant over time (no hourly profile), so temporal variability in tracer concentration for a given time and location is simply due to dispersion, and therefore to the meteorological variability. 335 This is why we use two sets of meteorological input data (*i.e.* NCEP–FNL and ECMWF–ERA5) driving the dispersion simulated with WRF–CHIMERE. Then we use the average of the tracer concentrations obtained for the two dispersion simulations run by NCEP–FNL and ECMWF–ERA5 for each city.

3. The concentration of the tracers emitted from each city is modeled independently. It corresponds to an arbitrary unit (a.u.). Nevertheless, in the model, there is a given emitted quantity (of 1000 tons per day) at each source point (*i.e.* 340 city center), and the a.u. is equivalent to ppb with this emitted quantity of tracers. The tracers are removed by transport outside of the domain or by deposition.

4. The tracer concentration from each source point is interpolated along the flight paths, as it is done for the pollutants studied. The location of city plumes is identified by retaining only those flight legs corresponding to a tracer concentration above a given threshold. Consequently the legs corresponding to city plumes are those where the tracer concentration of 345 at least one city is above the threshold, which is referred to as maximum of city tracer concentration.

For the two selected flights (*cf.* Section 3.1), the different city tracer concentrations are presented for both NCEP–FNL and ECMWF–ERA5 (Figure A14 and Figure A15). The city tracer concentrations are compared to the observed concentrations of the studied pollutants. The identification of the city plumes is done for three different thresholds of tracer concentration



covering two orders of magnitude (*i.e.* 0.1, 1 and 10 a.u.). The identification is shown for the two previously selected flights

(Figure 7 and Figure 8).

– "English Channel Flight" - 17 July 2017

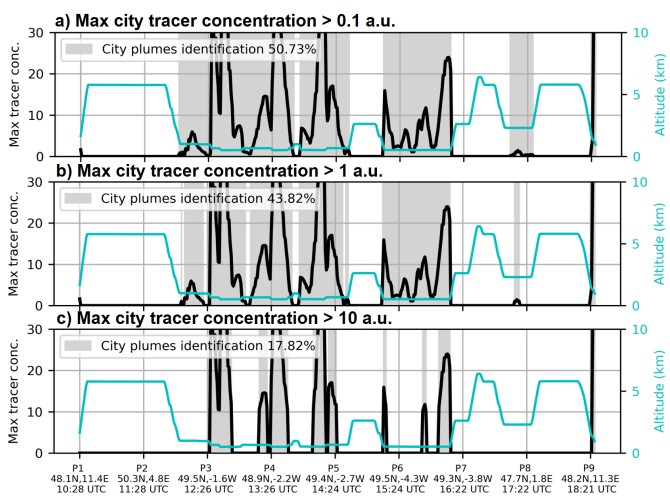

**Figure 7.** *City plume identification of the "English Channel Flight" occurring on 17 July 2017 using a methodology based on tracers emitted at selected city centers and transported using the dispersion simulated with WRF–CHIMERE. Flight legs corresponding to city plumes are those where the tracer concentration of at least one city (Max city tracer concentration) is above a threshold. The max city tracer concentration is modeled at each location and time step of the aircraft (black line) and the corresponding altitude is displayed (light blue line). Three thresholds of city tracer concentration are used, covering two orders of magnitude a) 0.1, b) 1 and c) 10 a.u. (arbitrary unit).*

For the "English Channel Flight" (Figure A14), we see that the tracers associated with London and, to a lesser extent, Manchester have concentration greater than 10 a.u. between P3 and P5, then their concentrations drop sharply to less than 1 a.u. between P5 and P6, and increase again greater than 10 a.u. between P6 and P7. The variability of the concentration of the

London tracers corresponds very well to the variability observed for BC and OA, as well as for CO, HCHO, $NO_2$ and $SO_2$ (*cf.* Figure 3 and Figure 2). Moreover, it can be seen that the Paris tracer concentrations reach 1 a.u. between P7 and P8. The percentage of time in city plumes decreases by a factor of 2.8 when increasing the threshold from 0.1 to 10 a.u. (Figure 7), which shows that the city plumes are clearly located because the maximum of city tracer concentration increases by more than two orders of magnitude.

– "Manila Flight" - 28 March 2018

For the "Manila Flight" (Figure A15), the Manila tracer concentration increases three times (greater than 10 a.u.), which also corresponds very well to the variability observed for BC and OA, as well as for CO, HCHO, $NO_2$ and $SO_2$ (*cf.* Figure 5 and Figure 4). From P6 to the end of the flight, the various locations of tracers released along the west coast of the island of





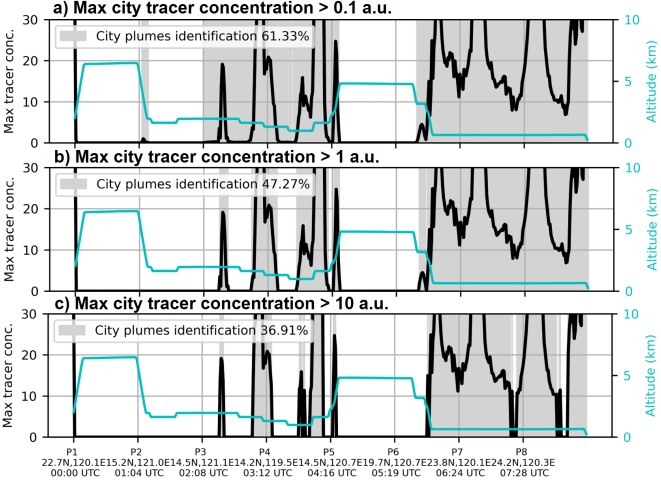

**Figure 8.** *City plume identification of the "Manila Flight" occurring on 28 March 2018 using a methodology based on tracers emitted at selected city centers and transported using the dispersion simulated with WRF–CHIMERE. Flight legs corresponding to city plumes are those where the tracer concentration of at least one city (Max city tracer concentration) is above a threshold. The max city tracer concentration is modeled at each location and time step of the aircraft (black line) and the corresponding altitude is displayed (light blue line). Three thresholds of city tracer concentration are used, covering two orders of magnitude a) 0.1, b) 1 and c) 10 a.u. (arbitrary unit).*

Taiwan are associated with high tracer concentrations (greater than 10 a.u.) as expected due to the flight plan. It should be noted
that the tracers emitted along the east coast of China are associated with non negligible tracer concentrations (greater than 0.1 a.u.), which shows the importance of the long-range transport of pollutants from China. The percentage of time in city plumes decreases by a factor of 1.7 when increasing the tracer concentration threshold from 0.1 to 10 a.u. (Figure 8), highlighting successful plume identification.

The choice of the threshold value has a small influence because the tracer concentration increases by more than two orders
of magnitude when the aircraft is located in city plumes. For the targeted megacities (London, Paris, Manila, Taipei), the tracer concentration of five source points leads to the same location of city plumes in both time and space, even after traveling several hundred kilometers (Figure A14 and Figure A15). Moreover, the location of city plumes is the same for both meteorological input datasets, which reinforce our confidence in the methodology to identify flights legs associated to city plumes.

## 4.2 Model ensemble performance in city plumes

From the identification of the plumes made with the tracer methodology, we evaluate the air quality model ensemble specifically in city plumes as it was done for various environments (*cf.* Section 3.2). We compare the R-values obtained between the modeled and observed variables using the three tracer concentration thresholds to understand their influence on the identification of city plumes (Table 4).



**Table 4.** *Correlation coefficients (R) between observed and modeled variables by an air quality model ensemble of the four simulations for wind speed, the five studied trace gases (CO, HCHO, NO₂, O₃, and SO₂), as well as black carbon (BC) and organic aerosol (OA). The R-values at a 1-min averaging time step are shown for the observations of the two EMeRGe campaigns that corresponds to city plumes as identified with three different intensities (Thresholds of 0.1, 1 and 10 a.u.).*

| Correlation coefficient (R) | | | |
|---|---|---|---|
| **Model** | CAMchem CESM2 | CAMS forecast | WRFchem FNL | WRFchem ERA5 |
| **EMeRGe - City plumes - Threshold 0.1 a.u. - N = 3407** | | | | |
| Wind speed | 0.58 | 0.88 | 0.90 | 0.91 |
| CO | 0.66 | 0.83 | 0.79 | 0.78 |
| HCHO | 0.44 | 0.51 | 0.53 | 0.52 |
| NO₂ | 0.35 | 0.47 | 0.53 | 0.63 |
| O₃ | 0.59 | 0.64 | 0.65 | 0.65 |
| SO₂ | 0.11 | 0.23 | 0.16 | 0.17 |
| BC | 0.44 | 0.35 | 0.57 | 0.59 |
| OA | 0.17 | -0.01 | 0.08 | 0.17 |
| **EMeRGe - City plumes - Threshold 1 a.u. - N = 2219** | | | | |
| Wind speed | 0.57 | 0.85 | 0.87 | 0.88 |
| CO | 0.64 | 0.79 | 0.75 | 0.75 |
| HCHO | 0.37 | 0.47 | 0.49 | 0.50 |
| NO₂ | 0.29 | 0.42 | 0.49 | 0.62 |
| O₃ | 0.57 | 0.62 | 0.65 | 0.65 |
| SO₂ | 0.14 | 0.23 | 0.15 | 0.16 |
| BC | 0.39 | 0.21 | 0.57 | 0.61 |
| OA | 0.26 | -0.03 | 0.11 | 0.23 |
| **EMeRGe - City plumes - Threshold 10 a.u. - N = 1010** | | | | |
| Wind speed | 0.60 | 0.82 | 0.84 | 0.85 |
| CO | 0.50 | 0.65 | 0.62 | 0.62 |
| HCHO | 0.21 | 0.28 | 0.32 | 0.38 |
| NO₂ | 0.37 | 0.38 | 0.43 | 0.58 |
| O₃ | 0.44 | 0.61 | 0.56 | 0.55 |
| SO₂ | 0.19 | 0.24 | 0.12 | 0.15 |
| BC | 0.16 | 0.16 | 0.50 | 0.45 |
| OA | 0.21 | 0.04 | -0.03 | 0.27 |



By increasing the value of the tracer concentration thresholds (*i.e.* the pollution intensity in city plumes), the R-values are slightly decreased, so there is little influence of this choice. A threshold of 1 a.u. is suitable to identify flight legs associated with high urban pollution, which corresponds to 32% of the two campaigns (while it is 48% for a threshold of 0.1 a.u. and 14% for a threshold of 10 a.u.). We focus in the following on a threshold of 1 a.u. to identify the flight legs in city plumes sampled during the campaigns.

The performance in city plumes is generally better for regional simulations compared to global simulations, especially for HCHO, $NO_2$, and BC. In city plumes, we can see that the R-values for BC are similar for the two regional simulations (*i.e.* WRFchem–FNL and WRFchem–ERA5), while for OA the R-value is lower for WRFchem–FNL than for WRFchem–ERA5. Comparing this evaluation of the model ensemble focused on city plumes with the evaluation for all observations (comparing Table 4 with Table 3), the performance of the model ensemble is poorer in city plumes compared to all environments and especially for OA.

Linear regressions between the observations and the four simulations are compared for all observations of the two campaigns and for observations in city plumes (Figure A6 for BC, Figure A7 for OA, Figure A8 for CO, Figure A9 for HCHO, Figure A10 for $NO_2$, Figure A11 for $O_3$, Figure A12 for $SO_2$, and Figure A13 for wind speed). While the R-values decrease significantly for two global simulations and to a lesser extent for the regional simulations, the slopes of the linear regressions remain mostly similar in city plumes (compared to the slopes obtained for all observations of the two campaigns). This is the case for all pollutants except for $O_3$, which is remarkable because the R-values remain mostly similar.

Focusing on city plumes, the statistical evaluation of the air quality model ensemble is degraded with respect to all observations and especially for the two global simulations. The model ensemble has moderate agreement for BC in city plumes (R ≈ 0.5), while there is no agreement anymore for OA (R < 0.2), which may be related to the production of SOA. Therefore, the relative amounts of carbonaceous aerosols and trace gases may not be adequately modeled, especially in city plumes.

## 4.3 Concentration ranges in city plumes

By focusing on city plumes, we notice, as expected, that the observed concentrations increase significantly and with different fractions depending on the variables (Figure 9 and Figure A16).

The carbonaceous aerosol observations show a decrease in the fraction of low concentrations (from 58 % to 46 % for BC and from 36 % to 7 % for OA), while the fraction of high concentrations (greater than 1 $\mu$g/m$^3$) increases significantly for BC and OA. The air quality model ensemble reproduces a decrease in the fraction of low concentrations for BC except for the CAMS–forecast, whereas for OA only the CAMS–forecast reproduces it. The model ensemble overestimates the fraction of high concentrations for BC and OA. For CO, the observed concentration range is similar when focusing on city plumes, while for $O_3$ there is an increase in high concentrations above 80 ppb. The fractions of low concentrations for CO and $O_3$ are overestimated by the model ensemble in city plumes, except for the CAMS–forecast. The fraction of high $O_3$ concentration is not reproduced by the model ensemble. For both HCHO and $NO_2$, the fractions of observed low concentrations are reduced (corresponding to concentrations less than 0.5 ppb and 0.2 ppb, respectively), which is reproduced by the model ensemble for HCHO but not for $NO_2$. Conversely, the fraction of observed high concentrations is reproduced for $NO_2$ but not for HCHO.





**Figure 9.** *Observed and modeled concentration ranges for all observations and in city plumes of (a) BC, (b) OA, (c) CO and (d) $O_3$ for the EMeRGe campaigns in Europe and in Asia at 1-min averaging time step.*

For $SO_2$, the modeled overestimation of the fraction of observed high concentration is increased in city plumes compared to all observations.

In conclusion, the overall performance is decreased in city plumes because the fractions of high concentration are overestimated for BC, OA, HCHO, and $SO_2$, which degrades the performance of the ensemble. The decrease in statistical metrics is more pronounced for carbonaceous aerosols, HCHO, and $SO_2$, which we attribute primarily to inaccurate anthropogenic emissions rather than to the modeled chemistry or the identification of city plumes.



## 5    Conclusions and perspectives

This comprehensive analysis contributes to the improvement of air quality modeling by assessing the performance of an air quality model ensemble in city plumes. The assessment of the air quality model ensemble focuses on both carbonaceous aerosols and five trace gases during the two EMeRGe campaigns. These campaigns, designed with similar flight plans for Europe and Asia, along with identical instrumentation, provide a unique opportunity to evaluate air quality models, having a specific focus on city plumes.

An air quality model ensemble is used to represent the current state-of-the-art in atmospheric modeling. This ensemble includes two global models, CAMS–forecast and CAMchem–CESM2, along with two regional WRFchem simulations using meteorological input datasets from NCEP–FNL and ECMWF–ERA5. A statistical evaluation of the air quality model ensemble for the two campaigns reveals very good agreement with the observations for wind speed. Gaseous pollutants with lifetimes longer than a day, such as CO and $O_3$, are well represented, while pollutants with shorter lifetimes (HCHO, $NO_2$, $SO_2$) show

poorer agreement, partly due to the large concentration ranges of the studied environments.

The performance of the air quality model ensemble exhibits significant differences between regions when comparing the two campaigns, with CO better reproduced in Asia and other pollutants studied showing better agreement in Europe. The performance for BC is similar between regions, whereas for OA, it is better in Europe. In addition, the observed variability in the relative amounts between BC, OA and the five trace gases is not well reproduced by the air quality model ensemble,

suggesting that emission factors in the inventories may contribute to the lack of accurate representation of the relative amounts between pollutants. This aspect deserves further investigation.

The modeled wind speed is in very good agreement with the observations, supporting the use of the modeled pollutant transport to identify the flight legs associated with pollution plumes originating from major population centers. The identification of city plumes targeted by the different flight plans is obtained from a methodology based on numerical tracer experiments, for

which tracers are emitted from major population centers. The methodology shows robust capabilities to localize in time and space the city plumes that have traveled several hundred kilometers from the different cities sampled during both campaigns. Focusing on city plumes degrades the performance of the ensemble, while the observed concentration ranges for all pollutants are reproduced by the ensemble, although the fractions of high concentrations are overestimated for BC, OA, HCHO, and $SO_2$.

*Author contributions.*  Conceptualization: AD, BG, PT, MV, JPB,

Data curation: MDAH, SB, BH, KK, FK, OOK, ML, KP, MP, HS, JS, BW,

Investigation: AD, MV, JPB,

Methodology: AD,

Resources: MV, BG, GB, GS,

Validation: AD, MV, BG, BS, KV,

Visualization: AD, IL,

Writing – original draft preparation: AD,



Writing – review & editing: All co-authors contributed.

*Competing interests.* At least one of the (co-)authors is a member of the editorial board of Atmospheric Chemistry and Physics.

*Acknowledgements.*

Financial support:

Adrien Deroubaix acknowledge the European Union's Horizon 2020 research and innovation programme for supporting this work under the Marie Skłodowska-Curie grant agreement No 895803 (MACSECH — H2020-MSCA-IF-2019).

For the funding of the HALO aircraft and the contributions to the various missions via the German Research Foundation (DFG; HALO-SPP 1294), the Max Planck Society (MPI), the Helmholtz-Gemeinschaft, and the Deutsches Zentrum für Luft- und Raumfahrt (DLR; all 460 from Germany) are highly acknowledged.

This study was in part funded by the State and University of Bremen. The EMeRGe study in Bremen was funded by the DFG Project number 316834290, which is a sub project of the DFG SPP 1294: Atmospheric and Earth System Research with the "High Altitude and Long Range Research Aircraft" (HALO).

The scientific work of Flora Kluge, Klaus Pfeilsticker, and Benjamin Weyland has been supported by the German Research Foundation 465 (DFG; grant nos. PF-384/7-1, PF384/9-1, PF-384/16-1, PF-384/17, PF-384/19, PF-384/24 and BU 2599/10-1).

Johannes Schneider, Katharina Kaiser, and Stephan Borrmann acknowledge funding through the DFG (project no. 316589531).

Acknowledgements:

The computation of the simulations presented in this work was completed by different supercomputers:

– For WRFchem, the authors gratefully acknowledge the resources granted by the Deutsches Klimarechenzentrum (DKRZ) granted by its Scientific Steering Committee (WLA) under project ID bb1260;

– For WRF–CHIMERE, the authors gratefully acknowledge the support provided by Pablo Echevarria and the computing time allocated on Hypatia at IUP.

Data availability:

– For the observational data, we thank the EMeRGe project for sharing the data, which are available through this website after registration: https://halo-db.pa.op.dlr.de/;

– For CAMS–forecast, data are available through this website:

https://ads.atmosphere.copernicus.eu/cdsapp#!/dataset/cams-global-atmospheric-composition-forecasts;

– For CAMchem–CESM2, data are available through this website: https://www.acom.ucar.edu/cam-chem/cam-chem.shtml.



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





 **Appendix A: Supplemental Material**

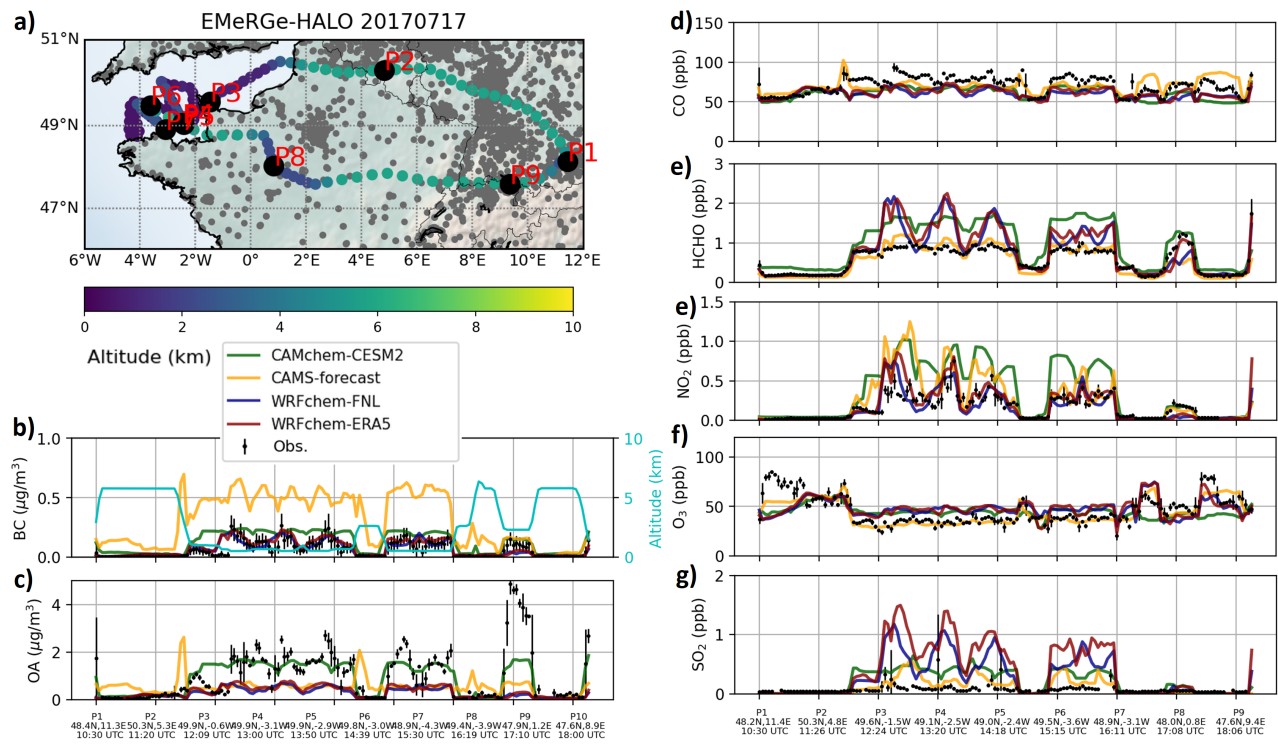

**Figure A1.** *a) Map of the EMeRGe flight on 17 July 2017, and time series of observed and modeled concentrations at a 3-min averaging time step of b) black carbon (BC) and (c) organic aerosol (OA), d) carbon monoxide (CO), e) formaldehyde (HCHO), f) nitrogen dioxide (NO₂), g) ozone (O₃) and h) sulfur dioxide (SO₂). Observations are presented with the standard deviation of the measurements during the time step (black dots with vertical bars). The air quality model ensemble (colored lines) is composed of two global simulations, CAMchem–CESM2 and CAMS–forecast and two regional simulations, WRFchem–FNL and WRFchem–ERA5. Aerosol concentrations are measured with a cut-off diameter of 1 μm and the modeled concentrations are shown accordingly.*





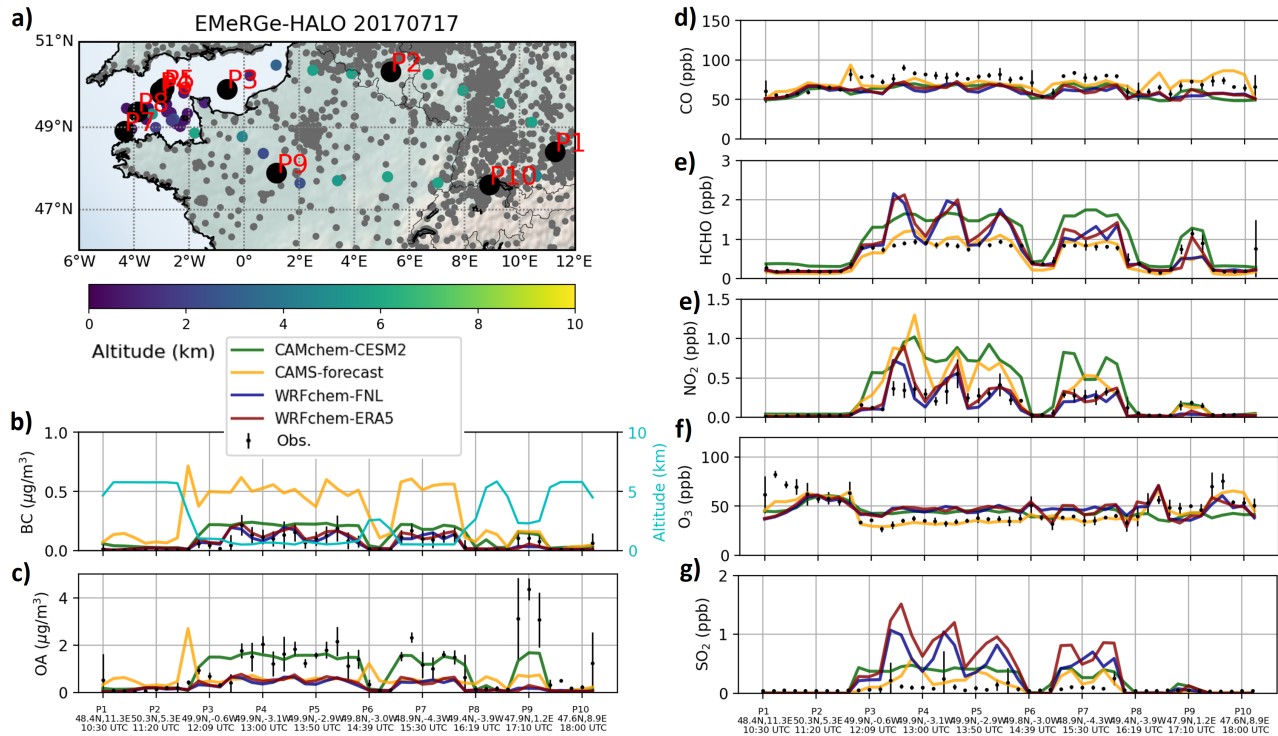

**Figure A2.** *a) Map of the EMeRGe flight on 17 July 2017, and time series of observed and modeled concentrations at a 10-min averaging time step of b) black carbon (BC) and (c) organic aerosol (OA), d) carbon monoxide (CO), e) formaldehyde (HCHO), f) nitrogen dioxide (NO$_2$), g) ozone (O$_3$) and h) sulfur dioxide (SO$_2$). Observations are presented with the standard deviation of the measurements during the time step (black dots with vertical bars). The air quality model ensemble (colored lines) is composed of two global simulations, CAMchem–CESM2 and CAMS–forecast and two regional simulations, WRFchem–FNL and WRFchem–ERA5. Aerosol concentrations are measured with a cut-off diameter of 1 μm and the modeled concentrations are shown accordingly.*




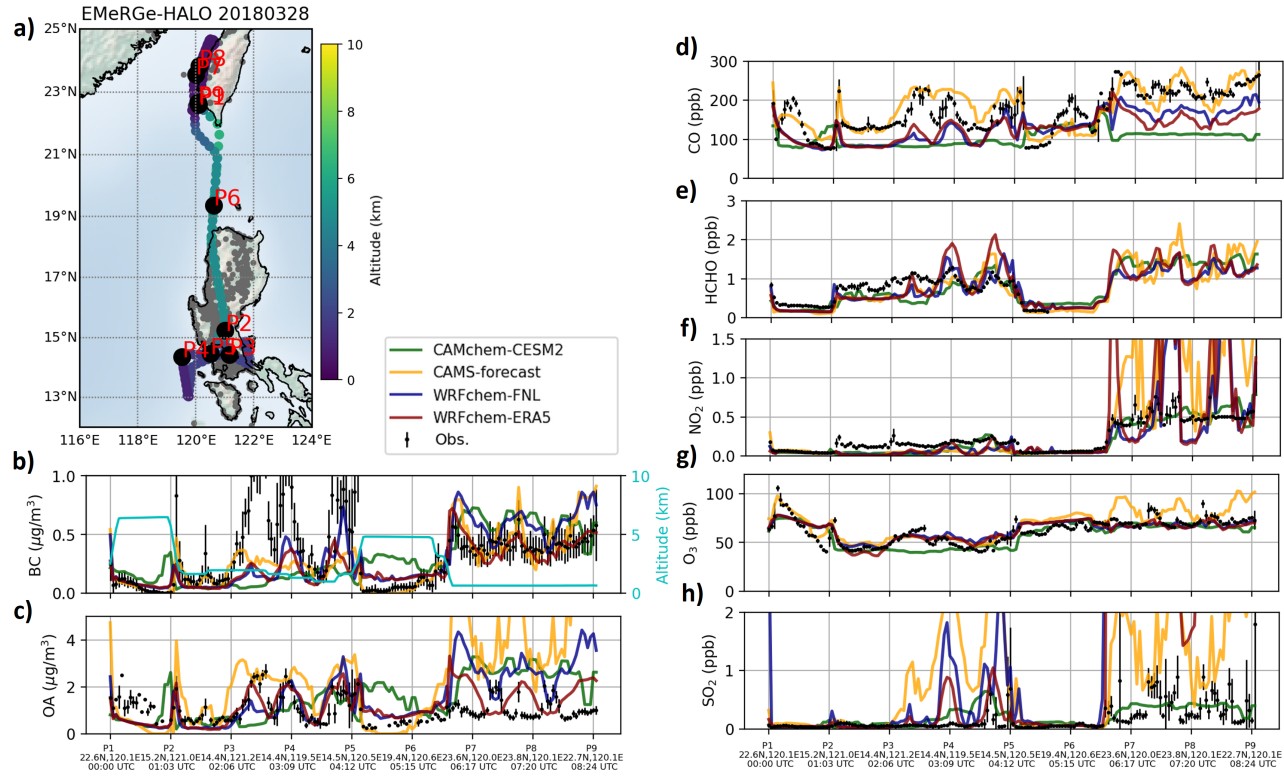

**Figure A3.** *a) Map of the EMeRGe flight on 28 March 2018, and time series of observed and modeled concentrations at a 3-min averaging time step of b) black carbon (BC) and (c) organic aerosol (OA), d) carbon monoxide (CO), e) formaldehyde (HCHO), f) nitrogen dioxide (NO$_2$), g) ozone (O$_3$) and h) sulfur dioxide (SO$_2$). Observations are presented with the standard deviation of the measurements during the time step (black dots with vertical bars). The air quality model ensemble (colored lines) is composed of two global simulations, CAMchem–CESM2 and CAMS–forecast and two regional simulations, WRFchem–FNL and WRFchem–ERA5. Aerosol concentrations are measured with a cut-off diameter of 1 μm and the modeled concentrations are shown accordingly.*





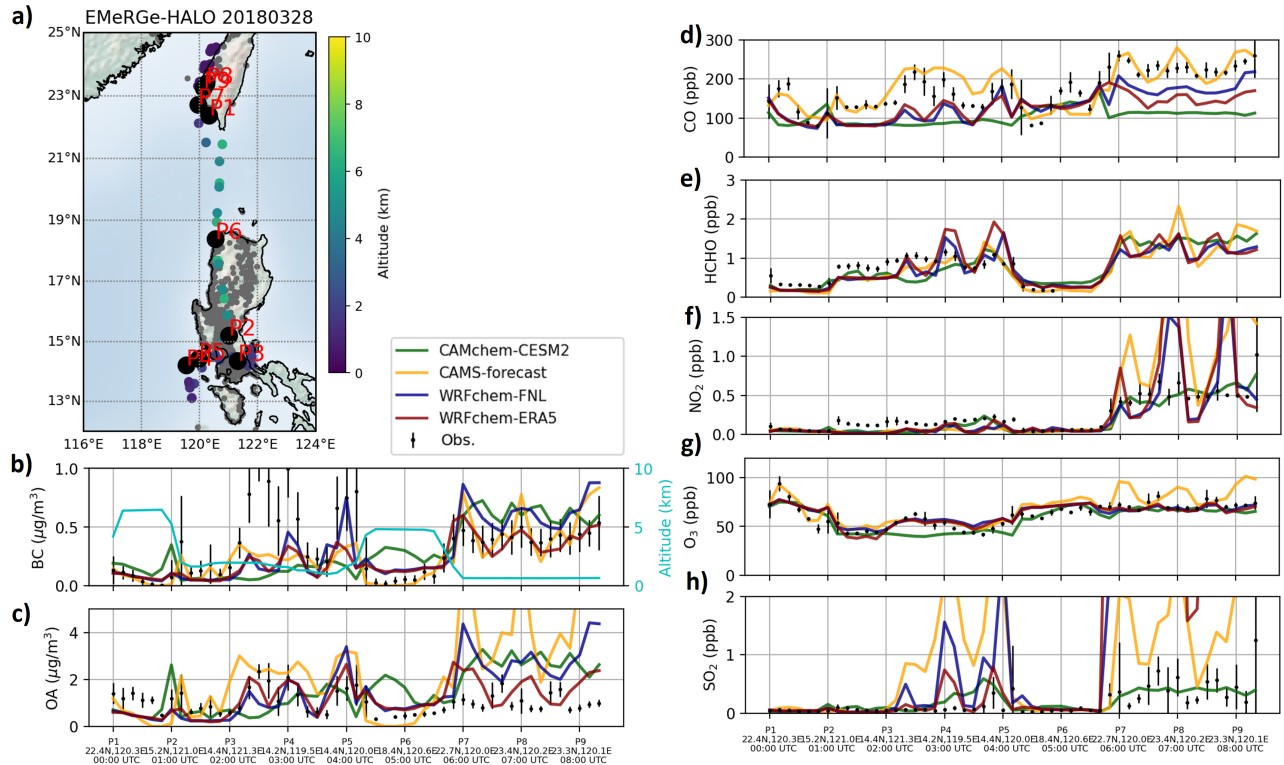

**Figure A4.** *a) Map of the EMeRGe flight on 28 March 2018, and time series of observed and modeled concentrations at a 10-min averaging time step of b) black carbon (BC) and (c) organic aerosol (OA), d) carbon monoxide (CO), e) formaldehyde (HCHO), f) nitrogen dioxide (NO$_2$), g) ozone (O$_3$) and h) sulfur dioxide (SO$_2$). Observations are presented with the standard deviation of the measurements during the time step (black dots with vertical bars). The air quality model ensemble (colored lines) is composed of two global simulations, CAMchem–CESM2 and CAMS–forecast and two regional simulations, WRFchem–FNL and WRFchem–ERA5. Aerosol concentrations are measured with a cut-off diameter of 1 μm and the modeled concentrations are shown accordingly.*





**Figure A5.** *Observed and modeled concentration ranges of (a) BC, (b) OA, (c) CO and (d) O$_3$ for the two campaigns.*



**Figure A6.** *Scatter plots of observed concentrations compared to that modeled interpolated along the flight positions for the two EMeRGe campaigns for BC at a 1-min averaging time step, a) in Europe, b) in Asia, c) for all observations of the two campaigns and d) in city plumes (determined with a threshold of 1 arbitrary unit of city tracer concentration). The air quality model ensemble is composed of two global forecasts, CAMchem–CESM2 (green line) and CAMS–forecast (orange line) and two regional simulations, WRFchem–FNL (blue line) and WRFchem–ERA5 (red line). Statistics associated to the regression lines (using reduced major axis regression) are given at the top for each simulation.*





**Figure A7.** *Scatter plots of observed concentrations compared to that modeled interpolated along the flight positions for the two EMeRGe campaigns for OA at a 1-min averaging time step, a) in Europe, b) in Asia, c) for all observations of the two campaigns and d) in city plumes (determined with a threshold of 1 arbitrary unit of city tracer concentration). The air quality model ensemble is composed of two global forecasts, CAMchem–CESM2 (green line) and CAMS–forecast (orange line) and two regional simulations, WRFchem–FNL (blue line) and WRFchem–ERA5 (red line). Statistics associated to the regression lines (using reduced major axis regression) are given at the top for each simulation.*



**Figure A8.** *Scatter plots of observed concentrations compared to that modeled interpolated along the flight positions for the two EMeRGe campaigns for CO at a 1-min averaging time step, a) in Europe, b) in Asia, c) for all observations of the two campaigns and d) in city plumes (determined with a threshold of 1 arbitrary unit of city tracer concentration). The air quality model ensemble is composed of two global forecasts, CAMchem–CESM2 (green line) and CAMS–forecast (orange line) and two regional simulations, WRFchem–FNL (blue line) and WRFchem–ERA5 (red line). Statistics associated to the regression lines (using reduced major axis regression) are given at the top for each simulation.*





**Figure A9.** *Scatter plots of observed concentrations compared to that modeled interpolated along the flight positions for the two EMeRGe campaigns for HCHO at a 1-min averaging time step, a) in Europe, b) in Asia, c) for all observations of the two campaigns and d) in city plumes (determined with a threshold of 1 arbitrary unit of city tracer concentration). The air quality model ensemble is composed of two global forecasts, CAMchem–CESM2 (green line) and CAMS–forecast (orange line) and two regional simulations, WRFchem–FNL (blue line) and WRFchem–ERA5 (red line). Statistics associated to the regression lines (using reduced major axis regression) are given at the top for each simulation.*



**Figure A10.** *Scatter plots of observed concentrations compared to that modeled interpolated along the flight positions for the two EMeRGe campaigns for NO$_2$ at a 1-min averaging time step, a) in Europe, b) in Asia, c) for all observations of the two campaigns and d) in city plumes (determined with a threshold of 1 arbitrary unit of city tracer concentration). The air quality model ensemble is composed of two global forecasts, CAMchem–CESM2 (green line) and CAMS–forecast (orange line) and two regional simulations, WRFchem–FNL (blue line) and WRFchem–ERA5 (red line). Statistics associated to the regression lines (using reduced major axis regression) are given at the top for each simulation.*





**Figure A11.** *Scatter plots of observed concentrations compared to that modeled interpolated along the flight positions for the two EMeRGe campaigns for O$_3$ at a 1-min averaging time step, a) in Europe, b) in Asia, c) for all observations of the two campaigns and d) in city plumes (determined with a threshold of 1 arbitrary unit of city tracer concentration). The air quality model ensemble is composed of two global forecasts, CAMchem–CESM2 (green line) and CAMS–forecast (orange line) and two regional simulations, WRFchem–FNL (blue line) and WRFchem–ERA5 (red line). Statistics associated to the regression lines (using reduced major axis regression) are given at the top for each simulation.*





**Figure A12.** *Scatter plots of observed concentrations compared to that modeled interpolated along the flight positions for the two EMeRGe campaigns for $SO_2$ at a 1-min averaging time step, a) in Europe, b) in Asia, c) for all observations of the two campaigns and d) in city plumes (determined with a threshold of 1 arbitrary unit of city tracer concentration). The air quality model ensemble is composed of two global forecasts, CAMchem–CESM2 (green line) and CAMS–forecast (orange line) and two regional simulations, WRFchem–FNL (blue line) and WRFchem–ERA5 (red line). Statistics associated to the regression lines (using reduced major axis regression) are given at the top for each simulation.*





**Figure A13.** *Scatter plots of observed concentrations compared to that modeled interpolated along the flight positions for the two EMeRGe campaigns for wind speed at a 1-min averaging time step, a) in Europe, b) in Asia, c) for all observations of the two campaigns and d) in city plumes (determined with a threshold of 1 arbitrary unit of city tracer concentration). The air quality model ensemble is composed of two global forecasts, CAMchem–CESM2 (green line) and CAMS–forecast (orange line) and two regional simulations, WRFchem–FNL (blue line) and WRFchem–ERA5 (red line). Statistics associated to the regression lines (using reduced major axis regression) are given at the top for each simulation.*



**Table A1.** *Correlation coefficients (R) between observed and modeled variables by an air quality model ensemble of the four simulations for wind speed, the five studied trace gases (CO, HCHO, NO₂, O₃, and SO₂), as well as black carbon (BC) and organic aerosol (OA). The R-values at 1, 3 and 10-min averaging time steps are shown during the EMeRGe campaigns, with those from Europe on the left side of the table (left part) and those from Asia on the right side.*

| Model | CAMchem CESM2 | CAMS forecast | WRFchem FNL | WRFchem ERA5 | CAMchem CESM2 | CAMS forecast | WRFchem FNL | WRFchem ERA5 |
|---|---|---|---|---|---|---|---|---|
| | EMeRGe–Europe | | | | EMeRGe–Asia | | | |
| **R at 1-min averaging time step** | | | | | | | | |
| Wind speed | 0.82 | 0.94 | 0.95 | 0.95 | 0.59 | 0.94 | 0.95 | 0.95 |
| CO | 0.42 | 0.52 | 0.49 | 0.45 | 0.41 | 0.79 | 0.74 | 0.73 |
| HCHO | 0.80 | 0.84 | 0.84 | 0.82 | 0.58 | 0.58 | 0.68 | 0.64 |
| NO₂ | 0.46 | 0.49 | 0.69 | 0.65 | 0.45 | 0.56 | 0.59 | 0.70 |
| O₃ | 0.69 | 0.77 | 0.69 | 0.68 | 0.38 | 0.54 | 0.48 | 0.48 |
| SO₂ | 0.28 | 0.45 | 0.41 | 0.41 | 0.12 | 0.26 | 0.18 | 0.19 |
| BC | 0.58 | 0.47 | 0.63 | 0.59 | 0.36 | 0.46 | 0.60 | 0.61 |
| OA | 0.68 | 0.58 | 0.67 | 0.67 | 0.25 | 0.32 | 0.42 | 0.48 |
| **R at 3-min averaging time step** | | | | | | | | |
| Wind speed | 0.82 | 0.95 | 0.95 | 0.95 | 0.60 | 0.94 | 0.96 | 0.96 |
| CO | 0.46 | 0.57 | 0.54 | 0.49 | 0.43 | 0.81 | 0.77 | 0.75 |
| HCHO | 0.80 | 0.85 | 0.84 | 0.82 | 0.58 | 0.59 | 0.68 | 0.64 |
| NO₂ | 0.70 | 0.77 | 0.70 | 0.69 | 0.39 | 0.55 | 0.49 | 0.49 |
| O₃ | 0.46 | 0.50 | 0.69 | 0.65 | 0.49 | 0.55 | 0.58 | 0.71 |
| SO₂ | 0.27 | 0.48 | 0.44 | 0.44 | 0.11 | 0.28 | 0.20 | 0.20 |
| BC | 0.61 | 0.51 | 0.67 | 0.64 | 0.38 | 0.47 | 0.61 | 0.62 |
| OA | 0.70 | 0.61 | 0.69 | 0.69 | 0.28 | 0.38 | 0.47 | 0.54 |
| **R at 10-min averaging time step** | | | | | | | | |
| Wind speed | 0.83 | 0.96 | 0.96 | 0.96 | 0.64 | 0.94 | 0.96 | 0.96 |
| CO | 0.54 | 0.67 | 0.61 | 0.57 | 0.46 | 0.83 | 0.81 | 0.78 |
| HCHO | 0.79 | 0.83 | 0.82 | 0.80 | 0.57 | 0.59 | 0.67 | 0.63 |
| NO₂ | 0.72 | 0.79 | 0.71 | 0.71 | 0.42 | 0.57 | 0.49 | 0.48 |
| O₃ | 0.46 | 0.52 | 0.71 | 0.68 | 0.39 | 0.58 | 0.62 | 0.68 |
| SO₂ | 0.26 | 0.50 | 0.52 | 0.52 | 0.10 | 0.28 | 0.18 | 0.16 |
| BC | 0.70 | 0.60 | 0.72 | 0.71 | 0.39 | 0.47 | 0.64 | 0.63 |
| OA | 0.73 | 0.66 | 0.72 | 0.71 | 0.30 | 0.38 | 0.47 | 0.54 |



**Table A2.** *Mean bias (modeled minus observed) between observed and modeled variables by an air quality model ensemble of the four simulations for wind speed, the five studied trace gases (CO, HCHO, NO$_2$, O$_3$, and SO$_2$), as well as black carbon (BC) and organic aerosol (OA). The values of mean bias at 1, 3 and 10-min averaging time steps are shown during the EMeRGe campaigns, with those from Europe on the left side of the table (left part) and those from Asia on the right side.*

| Model | CAMchem CESM2 | CAMS forecast | WRFchem FNL | WRFchem ERA5 | CAMchem CESM2 | CAMS forecast | WRFchem FNL | WRFchem ERA5 |
|---|---|---|---|---|---|---|---|---|
| | **EMeRGe–Europe** | | | | **EMeRGe–Asia** | | | |
| **Mean bias at 1-min averaging time step** | | | | | | | | |
| Wind speed (m.s$^{-1}$) | -2.151 | -0.112 | -0.556 | -0.573 | -2.262 | -0.876 | -0.895 | -1.185 |
| CO (ppb) | -21.538 | -3.897 | -22.281 | -22.022 | -63.151 | 9.564 | -48.140 | -49.118 |
| HCHO (ppbv) | 0.360 | -0.085 | 0.090 | 0.118 | -0.243 | -0.295 | -0.244 | -0.215 |
| NO$_2$ (ppb) | -0.026 | -0.059 | -0.099 | -0.094 | -0.042 | 0.232 | 0.119 | 0.345 |
| O$_3$ (ppb) | -3.262 | 2.305 | -1.084 | -0.778 | -2.409 | 4.599 | 0.029 | -0.849 |
| SO$_2$ (ppb) | -0.014 | -0.100 | 0.065 | 0.100 | 0.527 | 0.285 | 2.288 | 3.210 |
| BC ($\mu$g/m$^3$) | 0.058 | 0.175 | -0.010 | -0.009 | 0.428 | 0.150 | 0.075 | 0.057 |
| OA ($\mu$g/m$^3$) | 0.005 | -0.342 | -0.926 | -0.919 | 1.458 | 3.370 | 0.958 | 0.810 |
| **Mean bias at 3-min averaging time step** | | | | | | | | |
| Wind speed (m.s$^{-1}$) | -2.155 | -0.124 | -0.562 | -0.570 | -2.289 | -0.891 | -0.904 | -1.191 |
| CO (ppb) | -21.576 | -3.938 | -22.410 | -22.130 | -62.438 | 9.788 | -48.206 | -49.549 |
| HCHO (ppbv) | 0.356 | -0.087 | 0.080 | 0.108 | -0.243 | -0.297 | -0.245 | -0.217 |
| NO$_2$ (ppb) | -0.026 | -0.060 | -0.100 | -0.095 | -0.049 | 0.218 | 0.098 | 0.315 |
| O$_3$ (ppb) | -3.309 | 2.362 | -1.086 | -0.781 | -2.396 | 4.593 | 0.087 | -0.790 |
| SO$_2$ (ppb) | -0.012 | -0.101 | 0.059 | 0.097 | 0.527 | 0.259 | 2.218 | 3.158 |
| BC ($\mu$g/m$^3$) | 0.058 | 0.176 | -0.010 | -0.009 | 0.430 | 0.150 | 0.074 | 0.056 |
| OA ($\mu$g/m$^3$) | 0.011 | -0.328 | -0.915 | -0.908 | 1.443 | 3.162 | 0.905 | 0.751 |
| **Mean bias at 10-min averaging time step** | | | | | | | | |
| Wind speed (m.s$^{-1}$) | -2.158 | -0.186 | -0.590 | -0.626 | -2.311 | -1.015 | -0.960 | -1.225 |
| CO (ppb) | -22.022 | -4.118 | -22.796 | -22.375 | -61.966 | 9.822 | -48.752 | -50.010 |
| HCHO (ppbv) | 0.330 | -0.095 | 0.060 | 0.096 | -0.241 | -0.290 | -0.252 | -0.218 |
| NO$_2$ (ppb) | -0.027 | -0.058 | -0.101 | -0.095 | -0.039 | 0.220 | 0.064 | 0.262 |
| O$_3$ (ppb) | -3.295 | 2.239 | -1.282 | -0.990 | -2.527 | 4.397 | -0.109 | -0.949 |
| SO$_2$ (ppb) | -0.010 | -0.101 | 0.051 | 0.091 | 0.469 | 0.208 | 2.125 | 3.082 |
| BC ($\mu$g/m$^3$) | 0.057 | 0.175 | -0.012 | -0.011 | 0.438 | 0.149 | 0.073 | 0.056 |
| OA ($\mu$g/m$^3$) | -0.021 | -0.336 | -0.907 | -0.898 | 1.455 | 3.006 | 0.818 | 0.683 |



**Table A3.** *Root Mean Square Error (RMSE) between observed and modeled variables by an air quality model ensemble of the four simulations for wind speed, the five studied trace gases (CO, HCHO, NO$_2$, O$_3$, and SO$_2$), as well as black carbon (BC) and organic aerosol (OA). The RMSE-values at 1, 3 and 10-min averaging time steps are shown during the EMeRGe campaigns, with those from Europe on the left side of the table (left part) and those from Asia on the right side.*

| Model | CAMchem CESM2 | CAMS forecast | WRFchem FNL | WRFchem ERA5 | CAMchem CESM2 | CAMS forecast | WRFchem FNL | WRFchem ERA5 |
|---|---|---|---|---|---|---|---|---|
| | **EMeRGe–Europe** | | | | **EMeRGe–Asia** | | | |
| **RMSE at 1-min averaging time step** | | | | | | | | |
| Wind speed (m.s$^{-1}$) | 3.212 | 1.508 | 1.489 | 1.487 | 3.772 | 1.723 | 1.593 | 1.731 |
| CO (ppb) | 22.361 | 10.218 | 22.619 | 22.451 | 72.538 | 31.184 | 52.838 | 54.442 |
| HCHO (ppbv) | 0.434 | 0.190 | 0.211 | 0.239 | 0.328 | 0.422 | 0.332 | 0.353 |
| NO$_2$ (ppb) | 0.140 | 0.129 | 0.120 | 0.121 | 0.160 | 0.340 | 0.286 | 0.493 |
| O$_3$ (ppb) | 8.799 | 7.876 | 8.487 | 8.637 | 10.071 | 9.061 | 8.147 | 8.362 |
| SO$_2$ (ppb) | 0.164 | 0.140 | 0.211 | 0.245 | 0.934 | 0.683 | 2.538 | 3.446 |
| BC ($\mu$g/m$^3$) | 0.087 | 0.196 | 0.049 | 0.049 | 0.557 | 0.269 | 0.209 | 0.203 |
| OA ($\mu$g/m$^3$) | 0.669 | 0.778 | 0.961 | 0.950 | 1.950 | 3.613 | 1.349 | 1.198 |
| **RMSE at 3-min averaging time step** | | | | | | | | |
| Wind speed (m.s$^{-1}$) | 3.176 | 1.433 | 1.420 | 1.417 | 3.748 | 1.678 | 1.543 | 1.683 |
| CO (ppb) | 22.355 | 10.013 | 22.705 | 22.523 | 71.583 | 30.319 | 52.387 | 54.119 |
| HCHO (ppbv) | 0.432 | 0.190 | 0.208 | 0.234 | 0.326 | 0.421 | 0.332 | 0.353 |
| NO$_2$ (ppb) | 0.138 | 0.127 | 0.119 | 0.119 | 0.154 | 0.326 | 0.268 | 0.465 |
| O$_3$ (ppb) | 8.581 | 7.745 | 8.327 | 8.487 | 9.868 | 8.859 | 7.898 | 8.137 |
| SO$_2$ (ppb) | 0.167 | 0.142 | 0.202 | 0.239 | 0.951 | 0.669 | 2.468 | 3.400 |
| BC ($\mu$g/m$^3$) | 0.085 | 0.195 | 0.047 | 0.047 | 0.557 | 0.266 | 0.207 | 0.201 |
| OA ($\mu$g/m$^3$) | 0.643 | 0.750 | 0.944 | 0.933 | 1.897 | 3.402 | 1.278 | 1.124 |
| **RMSE at 10-min averaging time step** | | | | | | | | |
| Wind speed (m.s$^{-1}$) | 3.093 | 1.283 | 1.296 | 1.304 | 3.657 | 1.656 | 1.500 | 1.626 |
| CO (ppb) | 22.590 | 9.864 | 23.000 | 22.679 | 70.581 | 28.777 | 52.128 | 53.868 |
| HCHO (ppbv) | 0.424 | 0.193 | 0.212 | 0.240 | 0.327 | 0.417 | 0.334 | 0.353 |
| NO$_2$ (ppb) | 0.139 | 0.128 | 0.116 | 0.117 | 0.156 | 0.323 | 0.247 | 0.416 |
| O$_3$ (ppb) | 8.138 | 7.202 | 8.007 | 8.059 | 9.538 | 8.526 | 7.583 | 7.868 |
| SO$_2$ (ppb) | 0.166 | 0.143 | 0.186 | 0.226 | 0.984 | 0.697 | 2.473 | 3.433 |
| BC ($\mu$g/m$^3$) | 0.080 | 0.190 | 0.044 | 0.045 | 0.561 | 0.265 | 0.203 | 0.197 |
| OA ($\mu$g/m$^3$) | 0.586 | 0.697 | 0.927 | 0.917 | 1.887 | 3.287 | 1.225 | 1.093 |



**Table A4.** *Name and location of the emission point of numerical gaseous tracers from selected city. For Paris and London, five locations are used.*

| Country | City | Longitude | Latitude |
|---|---|---|---|
| **EMeRGe - Europe** | | | |
| France | Paris C | 2.34 | 48.85 |
| France | Paris N | 2.34 | 48.95 |
| France | Paris S | 2.34 | 48.75 |
| France | Paris W | 2.24 | 48.85 |
| France | Paris E | 2.44 | 48.85 |
| France | Marseille | 5.37 | 43.32 |
| France | Le Havre | 0.15 | 49.47 |
| France | Lyon | 4.84 | 45.75 |
| United Kingdom | London C | -0.12 | 51.50 |
| United Kingdom | London N | -0.12 | 51.60 |
| United Kingdom | London S | -0.12 | 51.40 |
| United Kingdom | London W | -0.22 | 51.50 |
| United Kingdom | London E | -0.02 | 51.50 |
| Germany | Manchester | -2.23 | 53.46 |
| Germany | Munich | 11.58 | 48.13 |
| Germany | Cologne | 6.96 | 50.93 |
| Germany | Stuttgart | 9.17 | 48.77 |
| Italy | Milan | 9.18 | 45.46 |
| Italy | Genoa | 8.87 | 44.41 |
| Italy | Turin | 7.66 | 45.07 |
| Italy | Venice | 12.25 | 45.44 |
| Italy | Rome | 12.47 | 41.89 |
| Spain | Barcelona | 2.16 | 41.38 |
| Belgium | Brussels | 4.35 | 50.84 |
| Netherlanders | Rotterdam | 4.47 | 51.90 |



**Table A5.** *Name and location of the emission point of numerical gaseous tracers from selected city. For Taipei and Manila, five locations are used.*

| Country | City | Longitude | Latitude |
|---|---|---|---|
| **EMeRGe - Asia** | | | |
| Taiwan | Taipei | 121.50 | 25.06 |
| Taiwan | Taoyuan | 121.22 | 24.95 |
| Taiwan | Taichung | 120.68 | 24.16 |
| Taiwan | Tainan | 120.20 | 22.99 |
| Taiwan | Kaohsiung | 120.35 | 22.62 |
| Philippines | Manila C | 121.03 | 14.60 |
| Philippines | Manila W | 120.93 | 14.60 |
| Philippines | Manila N | 121.03 | 14.70 |
| Philippines | Manila E | 121.13 | 14.60 |
| Philippines | Manila S | 121.03 | 14.50 |
| China | Guangzhou | 113.29 | 23.11 |
| China | HongKong | 114.13 | 22.35 |
| China | Xiamen | 118.15 | 24.48 |
| China | Fuzhou | 119.33 | 26.04 |
| China | Wenzhou | 120.70 | 28.00 |
| China | Hangzhou | 120.18 | 30.28 |
| China | Shanghai | 121.48 | 31.18 |
| China | Nantong | 120.89 | 31.99 |
| Japon | Osaka | 135.50 | 34.65 |



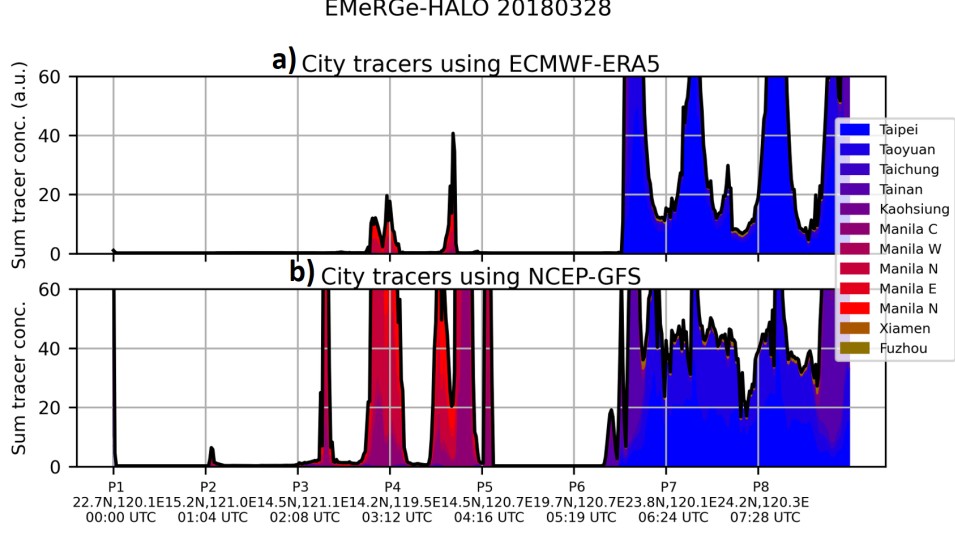

**Figure A14.** *Time series of the modeled tracer concentrations emitted at selected city centers (see color map) for their relevance to the trajectories of the HALO aircraft on 17 July 2017 at a 1-min averaging time step with the altitude (in km amsl), using two sets of input meteorological data, a) ECMWF–ERA5 and b) NCEP–FNL, to transport the tracers using the dispersion simulated with WRF–CHIMERE.*

**Figure A15.** *Time series of the modeled tracer concentrations emitted at selected city centers (see color map) for their relevance to the trajectories of the HALO aircraft on 28 March 2018 at a 1-min averaging time step with the altitude (in km amsl), using two sets of input meteorological data, a) ECMWF–ERA5 and b) NCEP–FNL, to transport the tracers using the dispersion simulated with WRF–CHIMERE.*



**Figure A16.** *Observed and modeled concentration ranges of (a) HCHO, (b) NO$_2$, (c) SO$_2$ and (d) wind speed for the EMeRGe campaigns in Europe and in Asia at 1-min averaging time step.*