# Peer review of "Air quality model assessment in city plumes of Europe and East Asia"

_EGUsphere, 2024_

## Author Comment (AC1)

**Revision egusphere-2024-516**

**To Editor:**

Dear Editor,

We thank the reviewers for their constructive comments and agree with their criticisms. The study presented is the first part of a project to study gas-aerosol relationships (funded by the European Commission through a Marie Skłodowska-Curie grant), which was not explained in the first version of the manuscript. The reviewers' comments showed us that it was a mistake not to refer to the overall objective of the project, as it is necessary to clearly define our research questions and justify the manuscript's focus on megacities and specific trace gases.

In this revised version, we clearly define the focus of our study on gas-aerosol relationships. The strength of observed gas-aerosol relationships from the EMeRGe campaign is now analyzed by comparing the performance of air quality models (through model-observation correlations) and gas-aerosol correlations in air quality modeling. Involving state-of-the-art air quality models should reveal strong gas-aerosol relationships observed, as they include all relevant processes currently known in a deterministic manner.

In the new version of the manuscript, we have changed the title, abstract, introduction, and conclusions of the article accordingly (see our replies to the reviewer comments below), and improved the readability of the manuscript. Moreover, the structure and figures have been largely revised.

Please find enclosed our point-by-point responses to the reviewers' comments, as well as a separate document with all the corrections to the manuscript, highlighted in blue for additions and red for removals.

We hope you will consider this revised manuscript for publication in ACP.

Yours sincerely
Adrien Deroubaix, on behalf of all co-authors

**RC1**

Deroubaix et al. present an evaluation of an air quality model ensemble using two aircraft campaigns focusing on carbonaceous aerosols and multiple trace gases. They find that each member of the ensemble reproduces carbon monoxide reasonably well, whereas the correlation between observation and the ensemble for organic aerosols is weak. Overall, the performed analysis is, however, superficial and focuses, at last in their discussion, on the Pearson correlation coefficient only. It includes many statements that are not supported by the analysis or are of a confusing manner. The abstract includes conclusions which are not discussed anywhere in the manuscript. The manuscript needs expanded revision to ensure a comprehensive analysis in order to meet the quality standards of ACP. Further, the manuscript needs major improvements in its language and presentation. Thus, I cannot support the publication of the manuscript in ACP.

We thank Reviewer 1 for their constructive criticisms, which have allowed us to substantially improve the manuscript. In this revised version, the objective of the study has been clearly redefined in the abstract and in the introduction. In addition, the emphasis on correlation coefficients is now justified as this serves as a measure of the strength of gas-aerosol relationships and of air quality model performance (Line 46-51 in the new version):

*The strength of gas-aerosol relationships in observations can be compared with the performance of air quality models, as well as with the gas-aerosol relationships reproduced by the models. In this study, we investigate gas-aerosol relationships from the EMeRGe database, and compare them with those simulated by air quality modeling. The Pearson correlation coefficient (R-value) is used to evaluate the strength of gas-aerosol relationships, as well as of the correlations between model and observation. The novelty lies in the evaluation of gas-aerosol relationships from observations and modeling in two different regions using identical aircraft instrumentation, with a special focus on city plumes.*

The manuscript has also been restructured and improved in terms of language and presentation to enhance the clarity of the message.

**Major comments**

Why is the analysis only performed for each individual ensemble member but not for the ensemble mean? The strength of a multi-model ensemble is to provide an estimate of the forecast uncertainty. Therefore, I would expect that the ensemble spread is discussed and evaluated with respect to the observation intercomparison. In the current version of the manuscript, the evaluation only considers a basic model intercomparison.

The objective and scientific questions have now been clearly defined in the revised version of the manuscript. In this context, we use an ensemble of state-of-the-art models to assess how well the models are able to represent observed pollutant concentrations, and the strength of correlations between observed aerosol and gas concentrations.

The mean of the model ensemble is useful for a large ensemble, and often yields a better forecast than any single model (*e.g Deroubaix et al., 2024, JGR*). However, the aim of this study is to use a few state-of-the-art air quality models representative of the ability of air quality modeling to investigate gas-aerosol relationships. Consequently, ensemble dispersion as a measure of prediction uncertainty is not discussed here.

A sentence has been added in Section 2.2 to express this idea (L 112-117 in the new version):

*The simulations of CAMchem--CESM2, CAMS--forecast and WRFchem are largely used in model intercomparison and ensemble forecasting systems (Solazzo et al., 2017; Petersen et al., 2019; Park et al., 2021; Deroubaix et al., 2024). We assume that the differences and similarities in model configuration allow this ensemble of four simulations to provide a good insight into the ability of air quality modeling to reproduce the spatial and temporal variability of the gases and aerosols studied and the strongest gas-aerosol relationships.*

Some of the trace gases analyzed have a strong diurnal cycle. Even though the authors discuss the impact of averaging the observations by 1, 3, and 10 minutes, the model output frequency provided for at least one model is 6 hours. I suspect that the low model output frequency (6 hours) and the performed interpolation has a much stronger effect on the evaluation. How do you justify that a 6 hour output is sufficient? Why not obtain model output at a higher frequency?

For the CAMchem-CESM2 and CAMS-ECMWF models, we use the data provided by NCAR and ECMWF at a resolution of 6h and 3h respectively. It is therefore not possible to modify the resolution. For our own regional model simulations, we have chosen a higher temporal resolution of 1 hour. The pollutants we study have a strong diurnal cycle near the surface in the planetary boundary layer, mainly due to strong diurnal cycles in emissions, but this is not necessarily the case at higher altitudes.

It should be kept in mind that the averaging of observations is based on measurements from a moving aircraft. In other words, an averaging time step of 1 minute corresponds to a horizontal distance of around 10 km, and an averaging time step of 10 minutes corresponds to a horizontal distance of around 100 km, which is comparable to the spatial resolution of regional or global models. Despite the large difference in sampled area, our results show that model performance remains stable between time steps of 1 minute, 3 minutes or 10 minutes (see Section 3.2 of the revised manuscript).

The authors tend to attribute differences between the models only to the different emissions used (e.g., line 417 or 435), even though the models differ in the gas phase chemical mechanism and the representation of aerosols. Further, these statements are pure guesses and are not supported by any in depth analysis. What are the differences in the emission inventories?

We agree with the reviewer that the four simulations studied differ not only for the emissions, but also for the gas phase chemical mechanism and the representation of aerosols. These statements have been removed because we need to investigate the emission inventories as mentioned by the reviewer. This aspect is discussed in a second article (https://doi.org/10.5194/egusphere-2024-521).

The motivation of the authors to use the selected two flight campaigns bases on the similarity of the flight plans. Later in the manuscript, however, the authors state that the flights differ significantly (different seasons, different time periods spend over the ocean, etc.). This is not consistent.

We thank Reviewer 1 for pointing out this imprecision. The two campaigns in different parts of the world obviously differ in the environment, climate etc. However, the strength of the EMeRGe (Effect of Megacities on the Transport and Transformation of Pollutants on the Regional to Global Scales) project is that the two campaigns were carried out with the same instrumental payload and the same goal of sampling city plumes. This enables a comparison of city plumes vs. background concentrations in the two regions.

We added to the manuscript that the similarity of flight plans relates to the fact of targeting city plumes during all flights with the same instrumental payload (L 65-70 in the new version):

*Both campaigns focused on the study of transport and chemical processes occurring in city plumes, and so most flights were conducted in the lower troposphere to sample city plumes (Figure A1). As the aircraft carried the same instrumental payload during both campaigns, it is possible to apply similar analyses in both regions. Despite the common objective of sampling city plumes, the HALO aircraft flew over the ocean more frequently in East Asia than in Europe, due to flight restrictions over China. This difference in flight paths may play an important role in the interpretation of the air masses sampled.*

Why are you only focusing on wind speed? The wind direction is equally important in order to assess transport pattern. How well does each ensemble member perform with respect to wind direction?

The reviewer is right that wind speed and direction are equally important to assess transport patterns of pollutants. The city plume identification in Section 4.1 of the revised version shows that modeled and observed pollutant concentration increases match well, demonstrating the high accuracy of the modeled meteorology.

Here, our objective is to focus on gas-aerosol relationships rather than to evaluate the modeled meteorology and transport patterns. That is why wind speed has been removed, and we only analyze the performance for gas and aerosol concentrations in the revised version.

Line 9: This statement is confusing. It sounds as if only due to the ensemble, differences in the observed concentrations are revealed.

The abstract has been revised and this confusing statement has been removed.

Line 11: I cannot find any statement about biomass burning in the main manuscript, except a general statement in the introduction.

The statements about biomass burning are not relevant in this revised version of the manuscript and have been removed.

Figure 1 is unreadable.

This figure has been moved to the supplement (as new Figure A1). The intention of the figure is to give an overview of the locations and altitudes where air masses were sampled by the aircraft, not to follow precisely the paths of individual flights.

Line 229 to 231: This statement is confusing. I would suggest to focus on the correlation coefficient, root-mean-square error, and the standard deviation to assess the performance of each ensemble member. Here, the use of Taylor diagrams would significantly improve the value and readability of the manuscript.

We agree that Taylor diagrams are valuable plots to visualize the performance of multiple models. However, in our case we only have 4 models, and Taylor diagrams would have to be done for each pollutant, i.e. 7 variables, leading to the addition of many figures. Instead, model performance of each ensemble member is visualized in different ways, e.g. their concentration means (triangles in Figure 1), or the heat maps/correlation matrices (Figures 2 and 3).

**RC2**

This manuscript evaluates the performance of three models in reproducing measurements from two aircraft campaigns conducted in Europe and East Asia. The focus is on carbonaceous aerosols and five trace gases, assessing whether the models can well simulate these pollutants in city plumes. They found good agreement for CO, while weak one for organic aerosol.

This manuscript is more like a measurement-model comparison report. It does not specify the initial scientific questions it aims to address and lacks in-depth analysis. It needs to be thoroughly revised to align with the standards of ACP. Below, I outline a couple of major concerns. The introduction section lacks a comprehensive literature review on the current state of the models used to simulate city plumes. Which models are typically used (why did the authors choose these three models, especially the two global models, for city-plume evaluation)? How effectively have these models reproduced city plumes? What obstacles are encountered (e.g., resolution being too coarse to capture local chemical/physical conditions)? Why is evaluating these models in two distinct regions (Europe and East Asia) beneficial for a better understanding of city plume modeling? Detailed clarifications are needed to help readers grasp the scientific importance of

this work. Also, these clarifications must be supported by substantial literature rather than merely instinctive judgments.

We thank Reviewer 2 for their constructive criticism. We apologize and acknowledge that the objective of the study was not well explained. In this revised version, we clearly define the objective of our study on gas-aerosol relationships, which is now explained in the first sentence of the abstract and in the last paragraph of the introduction.

The strength of observed gas-aerosol relationships is analyzed by comparing them to the performance of air quality models (through model-observation correlations) and to gas-aerosol correlations in air quality modeling. The interest of the state-of-the-art air quality models is that they should represent the strong gas-aerosol relationships observed, as they include all relevant processes currently known in a deterministic manner. We have added a sentence in the introduction (L 37-39 in the new version):

*Since the state-of-the-art air quality models represent atmospheric composition deterministically, the strongest gas-aerosol relationships should be reproduced even if the modeled composition does not perfectly align with observations. However, this aspect has not been extensively explored using airborne observations.*

And also added in Section 2.2 describing the air quality model ensemble (L 112-117 in the new version):

*The simulations of CAMchem--CESM2, CAMS--forecast and WRFchem are largely used in model intercomparison and ensemble forecasting systems (Solazzo et al., 2017; Petersen et al., 2019; Park et al., 2021; Deroubaix et al., 2024). We assume that the differences and similarities in model configuration allow this ensemble of four simulations to provide a good insight into the ability of air quality modeling to reproduce the spatial and temporal variability of the gases and aerosols studied and the strongest gas-aerosol relationships.*

If one of the authors' goals is to conduct a model intercomparison, why do the authors use different emission inventories for different models (as listed in Table 1). This setting complicates the comparison, making it hard to determine the source of the differences. I encourage the authors to use consistent emission inventories.

We use four different model simulations which largely differ in their configurations, such as their spatial and temporal resolutions and their emission inventories. Our goal is not to identify the causes of specific differences between models, rather we consider these simulations as representative of the state of the art of atmospheric modeling to evaluate model performance in relation to the strengths of observed gas-aerosol relationships (see previous response).

An introduction detailing the differences in chemical mechanisms and aerosol schemes across the models is necessary, given that this work involves model intercomparison.

As we now clearly define the focus of our study on gas-aerosol relationships, the introduction has been thoroughly revised (cf. first response).

It is necessary for the authors to provide more information about their aircraft campaign measurements: how each species is observed and what the uncertainties or detection limits of each instrument are. These uncertainties should be taken into account in the model-observation comparison.

We use observational data sets from instruments onboard which have been treated by the expert research groups responsible for each dataset. Measurements are averaged to 1-minute time steps and values below detection limits are removed. These values are very small compared to the observed variability for each species in the two regions which were sampled. A detailed article is dealing with the campaign overview by Andrés Hernández et al. (2022), which is referred to in L 71.

The paper is all about presenting simple statistical metrics of model-observation comparison, without an in-depth analysis of why these metrics appear similar or differ across species and regions. Moreover, it is unclear whether many of the metric presentations are helpful (back to my 1st major comment). For example, on Page 9 Line 165, why is CAMchem-CESM2 the only model that fails to reproduce high ozone concentrations? On Page 9 Line 185, how do the authors determine whether the better agreement found in ERA5 in terms of OA is due to better meteorological input, or if it is coincidentally correct for the wrong reasons, considering the uncertainties in OA chemistry? On Page 16 Lines 300-302, how do the authors conclude that fire emissions are the reason why CO is better reproduced in Asia? There are many other similar instances throughout the paper. If the authors do believe these issues are worth discussing, they should provide deeper analysis to support their statements.

The comments of the reviewer are pertinent and have highlighted the need to more clearly define the objective of the study and completely revise the manuscript (cf. first response). In the introduction of the new version, a paragraph has been added to explain clearly the objective of the study (L 46-51 in the new version):

*The strength of gas-aerosol relationships in observations can be compared with the performance of air quality models, as well as with the gas-aerosol relationships reproduced by the models. In this study, we investigate gas-aerosol relationships from the EMeRGe database, and compare them with those reproduced by air quality modeling. The Pearson correlation coefficient (R-value) is used to evaluate the strength of gas-aerosol relationships, as well as of the correlations between model and observation. The novelty lies in the evaluation of gas-aerosol relationships from observations and modeling in two different regions using identical aircraft instrumentation, with a special focus on city plumes.*

Regarding writing and organization, the sections presenting statistical metrics could be significantly shortened, while the interpretation of these comparisons should be expanded.

The manuscript has been substantially revised, including changes in the structure and a significant shortening of the text.

**Minor comments:**

Page 6 Line 123-124, what is the spatial representativeness of aircraft observations?

The spatial representativeness of aircraft observations averaged at 1-min time steps depends on the wind and how the dominant wind direction is crossed by the aircraft, the speed of the aircraft, and the measurement time step. At an approximate horizontal speed of 600 km/h, a 1-min averaging time step is representative of approximately 10 km along the flight path. In the new version, this is now explained in the Section 2.2 (L 118-126 in the new version):

*Based on the horizontal, vertical and temporal resolution of the outputs, the simulations are interpolated in time and in space according to the flight path of the HALO aircraft. In other words, the modeled concentrations are interpolated along the flight positions with a triple interpolation (bilinear horizontally, linear vertically and linear temporally between two time steps), which enables modeled concentrations to be generated at the locations and time steps of the observations made in the HALO aircraft. Since the aircraft is moving at a horizontal speed of approximately 600 km/h (10 km/min), the duration of the averaging time step can affect the ability of the air quality model ensemble to reproduce the observed concentrations, especially because of the difference in horizontal resolution ranging from 10 km for the regional simulations of WRFchem to about 100 km for the global simulation of CAMchem-CESM2. Consequently, we analyze different averaging time step durations (1-min, 3-min and 10-min) as we can expect an influence on air quality model performance.*

Figure 3 and 5, a legend is needed.

Figures 3 and 5 have been revised in the new manuscript and moved to the Supplement (Figures A5 and A6).

**RC3**

**General Comments**

This study compares model predictions from four models with flight data captured over Europe and Asia. A statistical analysis is performed to evaluate model-measurement agreement of BC, OA, CO, HCHO, NO2, O3, SO2, and wind speed. The paper is framed such that the model ensemble and comparison with observations will inform model development. However, the paper presents a simple listing of R-values for model-measurement comparisons, without describing properties of the models or physical systems that they aim to represent. This study will need major revisions in order to be published in ACP. Either much more analysis is required, or the paper must be reframed as a model evaluation paper. In the latter case, the paper may not be best-suited for ACP.

We thank Reviewer 3 for their constructive criticisms, and we agree that the focus on model performance via correlation coefficient (R-value) was not judicious because the overarching goal of the study was not explained.

In this revised version, the strength of observed gas-aerosol relationships is now analyzed by comparing with both the performance of air quality models (through model-observation correlations) and gas-aerosol correlations in air quality modeling. A paragraph has been added to explain clearly the objective of the study in the introduction (L 46-51 in the new version):

*The strength of gas-aerosol relationships in observations can be compared with the performance of air quality models, as well as with the gas-aerosol relationships reproduced by the models. In this study, we investigate gas-aerosol relationships from the EMeRGe database, and compare them with those reproduced by air quality modeling. The Pearson correlation coefficient (R-value) is used to evaluate the strength of gas-aerosol relationships, as well as of the correlations between model and observation. The novelty lies in the evaluation of gas-aerosol relationships from observations and modeling in two different regions using identical aircraft instrumentation, with a special focus on city plumes.*

**Specific Comments**

1. In line 80, the authors state that the trace gases were chosen because they are readily observable by satellites. This statement doesn't seem to fit with the remainder of the paper, since satellite observations are not used. If satellite relevancy is a main driver of this study, then it should be mentioned in the Introduction and Conclusions.

We apologize that the explanation for the choice of these gases was not sufficient. The relevance of gases observable by satellite is now explained in the abstract and the introduction together with the overarching goal of our research (cf. first response), and referred to again in the conclusions. We modified the first sentences of the abstract to explain this idea (L 1-7 in the new version):

*Aerosols are often studied as a single entity due to the difficulty of measuring their chemical composition. Estimating aerosol composition from satellites could progress from exploiting gas-aerosol relationships. This study analyzes the observed relationships between carbonaceous aerosols, i.e. black carbon (BC) and organic aerosol (OA), and five satellite-observable trace gases (CO, HCHO, NO2, O3, and SO2) during the EMeRGe (Effect of Megacities on the transport and transformation of pollutants on Regional and Global scales) aircraft campaigns in Europe and East Asia, and compares these relationships with air quality model performance, focusing on city plumes.*

2. The authors should consider adding another column to Table 1 + another row for meteorology, so that each of the WRF simulations (FNL and ERA5) have their own columns. Since you describe 4 models in the ensemble, it would be helpful to see 4 models in the table.

A 4th column has been added to Table 1 giving the configurations of the simulations studied.

   3. It would be easier to interpret the similarities between aerosols and gases if Figure 2b-c was part of Figure 3.

The former Figures 2 and 3 are no longer part of the main manuscript but have been moved to the supplement. We follow the suggestion of Reviewer 3 to combine Figures 2 and 3, which is Figure A5 in the new version of the Supplement.

   4. Consider replacing Tables 2-3 with heat maps, where the axes remain the same (model vs. variable), but each square of the table is filled with a color that scales with the R-value, e.g., darker color is higher R and lighter color is lower R. This could make it easier to see differences between "good" and "bad" predictions.

We thank the reviewer for this suggestion. In the revised manuscript, we show R-values as heat maps for an easier visual overview of the agreement between variables. The same kind of visualization has been used to compare the model performance with gas-aerosol relationships (new Figures 2, 3, 5 and 6).

   5. Sections 3.2.1 and 3.2.2 compare R-values (and bias and RMSE to some extent) between the Europe and Asia campaigns, but do not explain why these differences/similarities may have occurred. Could it be caused by similarities/differences in wind speed, topography, emission sources, etc? Does model configuration and design have an impact and why? Section 3.3 also comments on the ability of the models to reproduce observed concentrations but does not give possible explanations for differences. Please expand on these sections.

We agree with the reviewer that these sections were too superficial in the previous version. The focus of the article has been changed in order to assess the strength of observed gas-aerosol relationships during the two campaigns with air quality model performance. This is clearly explained in the introduction of the Section 3 (L 128-131 in the new version):

*In this section, the concentration ranges of carbonaceous aerosols and trace gases measured during the two EMeRGe campaigns in Europe and Asia are compared with the four simulations of the air quality model ensemble (Section 3.1). The performance of the ensemble is evaluated for both campaigns (Section 3.2), and compared to the strength of the observed gas-aerosol relationships (Section 3.3).*

   6. The sentence spanning lines 357-359 does not make sense to me, please rephrase.

The sentence has been rephrased from:

*The percentage of time in city plumes decreases by a factor of 2.8 when increasing the threshold from 0.1 to 10 a.u. (Figure 7), which shows that the city plumes are clearly located because the maximum of city tracer concentration increases by more than two orders of magnitude.*

to  (L 298-299 in the new version):

*By increasing the tracer concentration threshold from 0.1 to 10 a.u. (Figure 4), the percentage of time in city plumes decreases only slightly, by a factor of 2.8 for the "English Channel flight" and 1.7 for the "Manila flight", showing that city plumes are clearly localized.*

7. Section 4 does not fully explain why the largest megacities use five source points of tracer gas. Does this improve plume detection (like in Figures 7-8) as opposed to only having tracer "released" from one point in the city center?

Five source points are used because the area of the megacities is larger than a 10 by 10 km model grid cell, so that tracer emissions are released from five grid cells corresponding to the megacity's area. We see that for the megacities, the tracer concentration from five source points leads to close locations of the city plume, even after traveling several hundred kilometers (Figure A7 and Figure A8).

Two sentences have been added to Section 4.1 to provide more explanations (L 246-249 in the new version):

*For the targeted megacities (London, Paris, Manila, Taipei), we release tracers in five source points instead of one: a first is located in the city center  and the others 10 km to the north, east, south and west in order to release the tracers from five grid cells corresponding to the megacity's area.*

And also (L 298-299 in the new version):

*For the targeted megacities (London, Paris, Manila, Taipei), we also see that the tracer concentration from five source points leads to close locations of the city plume, even after traveling several hundred kilometers. These results show that the methodology accurately identifies the flight legs associated with city plumes.*

8. Reiterating the same comment as above (comment #5) for Sections 4.2-4.3. Section 4.3 has a small amount of physical reasoning (e.g., increased concentrations in plumes due to higher emissions), but both of these sections could use significantly more physical insight.

We again agree with the reviewer that these sections were too superficial in the previous version. The manuscript has been largely improved with many references to analyze the strength of gas-aerosol relationships observed during the two campaigns as well as the air quality model performance, especially in Section 4.3, for instance L 353-369 in the new version):

*The strength of the BC-OA, CO-BC and O3-OA relationships observed in the city plumes is stronger than in all environments (cf. Section 3.3). Our results suggest that BC, OA and CO share the same emission sources in cities, which is in agreement with other studies investigating urban stations in terms of R-values for the BC-OA relationship (e.g., Krivácsy et al., 2001; Sahu et al., 2011) and for BC-CO relationship (e.g., Baumgardner et al., 2002; Spackman et al., 2008). The O3-OA relationship stands out in city plumes, with R-values reaching 0.8 in Europe and 0.6 in Asia (compared to 0.4 in all environments). The strong O3-OA relationship in city plumes may be*

*related to the oxidation of VOC, which produces secondary OA (e.g., Turpin and Huntzicker, 1991; Jimenez et al., 2009; McMeeking et al., 2011; Yoon et al., 2021).*

9. The paper focuses almost entirely on R-values. Why can this be chosen as the only (and "best") analytic of model accuracy? Consider putting more metrics in the main text.

We apologize that we did not explain the reason for the focus on R-values. In the introduction of the revised version, a paragraph has been added to explain clearly the objective of the study and the interest of R-values (L 46-51 in the new version):

*The strength of gas-aerosol relationships in observations can be compared with the performance of air quality models, as well as with the gas-aerosol relationships reproduced by the models. In this study, we investigate gas-aerosol relationships from the EMeRGe database, and compare them with those reproduced by air quality modeling. The Pearson correlation coefficient (R-value) is used to evaluate the strength of gas-aerosol relationships, as well as of the correlations between model and observation. The novelty lies in the evaluation of gas-aerosol relationships from observations and modeling in two different regions using identical aircraft instrumentation, with a special focus on city plumes.*

10. Lines 417-418 states "we attribute primarily to inaccurate anthropogenic emissions rather than to the modeled chemistry or the identification of city plumes". How did you determine this and how can you show this?

Reviewer 3 is right because this statement is not fully demonstrated by our results. This section (Section 4.2 of the new version) has been largely revised for clarity, and this statement has been removed.

11. The first sentence of the Conclusions says that this study "contributes to the improvement of air quality modeling". However, this study applies existing models rather than making model improvements. Please rephrase.

We agree that this sentence was not appropriate. It has been removed from the text. The manuscript has been thoroughly revised and the goals defined precisely. The abstract, introduction and conclusion have been changed accordingly.

12. The last sentence of the abstract is inconsistent with what the paper presents. Because aspects of the models (chemical mechanisms, transport models, gridding, nesting, boundary conditions, etc.) and physical properties of the plumes/emissions/meteorology/chemistry/etc. are not discussed in the paper, the paper does not give a complete assessment of how the models do and do not perform well, or make suggestions on how to improve the models.

Reviewer 3 is right; this sentence has been removed from the abstract. The revised abstract is now consistent with the scientific goals and conclusions of the paper.

Technical Corrections

1. In general, much of the phrasing is confusing and the grouping of paragraphs is unorganized. I recommend having a few more people involved in proofreading the manuscript before re-submitting.

The manuscript has been restructured and thoroughly revised for the clarity of language.

2. Line 26: "early warning of the health impacts" should be "early warning of health impacts"

This sentence has been removed from the abstract.

3. Combine the paragraph starting at line 105 with the paragraph before it.

The paragraphs have been combined.

4. Remake Figure 2A so that the red P* labels do not overlap.

Figure 2a, and the equivalent Figure 4a for the second flight, have been moved to the supplement. It is complicated to move manually move all label locations, but the latitude and longitude of each P* are given in the corresponding time series of the flights in Figures 2b/c and 4b/c (now Figures Figure A5 and Figure A6 in the supplement).

5. Line 142: define amsl

The definition (above mean sea level) has been added.

6. Table 2:
    1. Missing the red WRFchem ERA5 label on the left side
    2. In the caption, you don't need "(left part)" in the last sentence
    3. Replace "- N" with ": N" so that it looks less like a minus sign

This table has been replaced by a Heatmap, as suggested by reviewer 3. We took care to use concise captions and correct labels.

7. Make it more clear in the paragraph beginning in line 320 that the tracers are "released" in the model, not in actual city plumes.

We rephrased the first sentence of the paragraph to be clear that the tracers are released in the model (L 236-237 in the new version):

*The methodology involves releasing tracers in the model from selected source points (i.e. grid cells) corresponding to city centers, which are transported depending on the dispersion simulated by the meteorological model*